# PROFIT: A Specialized Optimizer for Deep Fine Tuning

Anirudh S Chakravarthy    Shuai Kyle Zheng    Xin Huang    Sachithra Hemachandra

Xiao Zhang    Yuning Chai    Zhao Chen

GM Cruise LLC

## Abstract

The fine-tuning of pre-trained models has become ubiquitous in generative AI, computer vision, and robotics. Although much attention has been paid to improving the efficiency of fine-tuning models, there has been less scholarship around fine-tuning specifically for improved model performance. To remedy this gap, we present PROFIT, one of the first optimizers designed to incrementally fine-tune converged models on new tasks and/or datasets. Unlike traditional optimizers such as SGD or Adam, which make minimal assumptions due to random initializations, PROFIT takes the properties of a converged model into account explicitly to regularize the optimization process. Employing a temporal gradient-orthogonalization process, PROFIT outperforms fine-tuning methods in various tasks, from image classification to multimodal language model training to large-scale motion prediction. Moreover, PROFIT is encapsulated as a modular optimizer, which makes it easy to integrate directly into any training pipeline with minimal engineering effort.

## 1 Introduction

Fine-tuning pre-trained models has become widely adopted in solving computer vision and robotics problems as well as in modern generative AI settings. As datasets and models increase in size, having to train a new model for every new application and setting quickly becomes intractable. Imagine, for example, the cost of having to train a new model every time an autonomous vehicle needs to operate in a new city, or every time a camera application needs to recognize a new type of object, or every time a custom LLM needs to update its knowledge cutoff. The shift towards fine-tuning within the deep learning community has also been accelerated by developments in large foundational models that were trained on vast quantities of data, such as CLIP Radford et al. [2021], DINO Caron et al. [2021], and open-source LLMs Touvron et al. [2023a]. Indeed, deep learning is steadily inching towards a new paradigm where very few practitioners are training models from scratch.

At the same time, fine-tuning a model comes with its own set of challenges. For one, models are known to readily forget old information when fine-tuned with new information, in a process known as catastrophic forgetting Goodfellow et al. [2013]. Various mitigation methods have been proposed Li and Hoiem [2017], but require data engineering and modifications to the model architecture. Common practice within fine-tuning still largely relies on training on new tasks/data with a smaller learning rate or with a frozen backbone (or both). Parameter-efficient fine-tuning methods such as LoRA Hu et al. [2022] introduce learnable adapters for transfer learning, but serve to improve fine-tuning efficiency rather than accuracy (Biderman et al. [2024]). We aim to improve the *accuracy* of fine-tuning *while keeping the process efficient by not introducing any additional parameters*.

A ubiquitous object within deep learning that is modular and abstracted away from the practitioner is the optimizer. Many AI models set the optimizer to popular standards like Adam  Kingma and Ba

39th Conference on Neural Information Processing Systems (NeurIPS 2025).

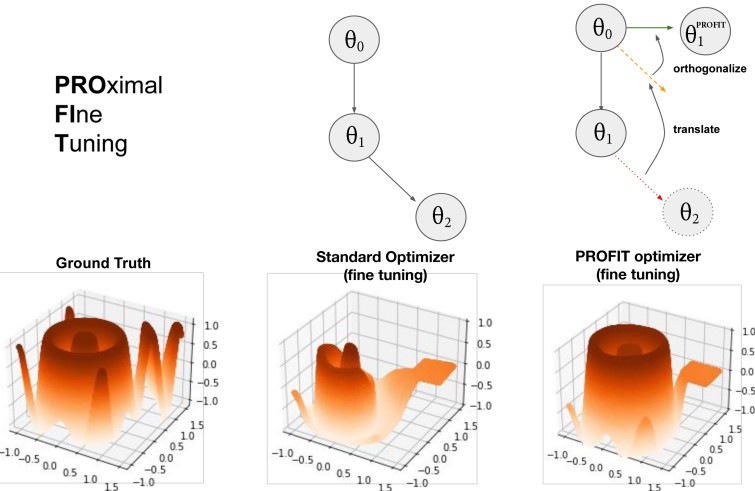

Figure 1: Schematic of PROFIT. Standard fine-tuning (middle) takes successive steps away from a good starting state $\theta_0$. PROFIT (right): (1) take $n_{\text{ref}}$ small reference steps with $O^{(\text{ref})}$ to obtain a displaced state $\theta_1$; (2) compute the displacement $\Delta = \theta_1 - \theta_0$; (3) orthogonalize the new-batch gradient $g := \theta_2 - \theta_1$ to $-\Delta$; (4) restore $\theta \leftarrow \theta_0$ and then apply the main optimizer $O$ along the orthogonalized direction. Here, r denotes 'reference'. See Alg. 1 for details.

[2015], AdamW Loshchilov and Hutter [2019], or Momentum Sutskever et al. [2013]. However, all current optimizers are designed for training from scratch, so they make minimal assumptions about the problem setting and initial model state. In contrast, the fine-tuning setting usually starts from a well-trained, well-converged model that already performs well on some set of meaningful data. So we ask: *how do we design an optimizer specifically to start from a converged model from a similar domain*? Given the ubiquity and modularity of optimizers within training pipelines, such an optimizer would be immediately applicable and easy to implement within any fine-tuning setting.

Works such as Learning Without Forgetting (LWF) Li and Hoiem [2017] have proposed mitigating catastrophic forgetting by enforcing proximity to an old state of the model, but require additional data pipelining and model snapshots that serve as additional supervision to keep the model anchored to its old "good" state. Instead of anchoring a model across tasks, we propose a different but equally valid anchoring *across time* rather than *across tasks*. For each iteration of the optimizer, we take a step away from equilibrium and *balance further steps away from equilibrium with the model's desire to return to equilibrium* using techniques from the gradient-based multi-task learning literature. The result is a system that mimics data-driven anchor methods such as Li and Hoiem [2017] without incurring any additional data processing, and which foregoes the rigid static anchors of more classic methods in favor of a dynamically updated flexible one.

Specifically, a model converged in some state $\theta_0$ will proceed along the states $\theta_1, \ldots \theta_t$ with a standard optimizer. A further update would send the system to $\theta_{t+1}$. However, because $\theta_0$ was a "good state," the model would also benefit by returning to $\theta_0$ (Sec. 3). We thus have two potentially conflicting gradient directions ($\boldsymbol{\Delta} := \theta_t \to \theta_0$ and $\mathbf{g} := \theta_t \to \theta_{t+1}$), which is a classic multitask learning problem. We borrow from Yu et al. [2020] and assign $\mathbf{g} \mapsto \mathbf{g} \perp \boldsymbol{\Delta}$, where $\perp$ is the orthogonalization operator. We restore the model to state $\theta_0$ ("translate" operation on the top right of Figure 1) and take a step in the orthogonalized direction $\mathbf{g}$.

The operation of PROFIT relies on some non-negligible distributional overlap between the fine-tuning and baseline settings, as it aims to dynamically keep the model state close to the baseline state. This requirement precludes some settings that are common within fine-tuning like pre-training and then fine-tuning on completely orthogonal datasets. We refer to our approach with this additional constraint as "proximal fine-tuning", where the fine-tuning dataset is of a similar distribution to that of the pre-training dataset. Although this constraint may seem limiting, "proximal fine-tuning" is prevalent in machine learning applications. For example, a self-driving car's trajectory prediction network may need to train a prediction head for a new scooter type using the same sensor, while

maintaining performance on old tasks. We will also later show how to overcome this constraint even in non-proximal settings by introducing a warmup phase.

PROFIT (PROximal FIne Tuning) is shown schematically in Fig. 1. To the best of our knowledge, PROFIT is among the first optimizers explicitly designed for fine-tuning.

Our main contributions are as follows.

- We introduce PROFIT, an optimizer for fine-tuning converged models, easily integrated into any deep learning framework.
- We show that PROFIT allows unsupervised training as if the original data were available.
- We show that PROFIT outperforms standard fine-tuning methods on various tasks, from image classification to VLM fine-tuning to large-scale motion prediction for autonomous driving.

## 2 Related work

**Multi-Task Learning**    For detailed background context in multi-task learning (MTL), we refer the reader to Zhang and Yang [2021]. MTL Zhang and Yang [2021] is an optimization problem in which we train a model on multiple tasks simultaneously to take advantage of the structure of shared neural networks, thus improving generalization and efficiency. One direction is to use gradient descent methods to optimize the joint multi-task learning problem. Our setting shares some similarities with MTL, with the key difference that problem is a temporal multi-task learning. Ozan Sener and Koltun [2018] formulates the MTL problem as a multi-objective optimization problem and learns the loss weights that change dynamically. GradNorm Chen et al. [2018] attempts to normalize gradients to balance learning of multiple tasks. PCGrad Yu et al. [2020] suggests that to mitigate gradient direction conflicts, we should project a task's gradient onto the normal plane of the gradient of any other task where a gradient conflict is present. Our proposed technique is similar to PCGrad, but modifies the core designs specifically for the "proximal fine-tuning" setting.

**Parameter-efficient fine-tuning**    Parameter-efficient fine-tuning reduces computational and memory requirements by updating fewer parameters, which simplifies the adaptation of large models to new tasks. An adapter Houlsby et al. [2019] introduces light-weight learnable parameters to help transfer learning. The trainable parameters in the adapter are much smaller than in the model, which makes them attractive in practice. Several strategies have been proposed, such as visual prompt tuning Jia et al. [2022], side adapters Zhang et al. [2020], bias tuning Cai et al. [2020], and residual adapters Rebuffi et al. [2017a] for efficient learning. LoRA Hu et al. [2022] decomposes the matrices of an attention module into a low-rank matrix. In contrast to our method, adapters require additional parameters and may also assume a specific class of model architectures for their success (e.g., transformers).

**Optimizer-based Fine-Tuning**    Fine-tuning on particular datasets Kornblith et al. [2019], Chen et al. [2020] is a common technique in the era of deep learning. Practitioners often use standard optimizers such as Stochastic Gradient Descent (SGD) Bottou [2010], Adam Kingma and Ba [2015], and AdamW Loshchilov and Hutter [2019]. Recent architectures such as ViTs Dosovitskiy et al. [2021], Caron et al. [2021], Radford et al. [2021] or ConvNeXts Liu et al. [2022] use AdamW for fine-tuning, while it is also common to use SGD for fine-tuning models like ResNets He et al. [2016], Kolesnikov et al. [2020] due to the optimizer's efficiency. However, SGD and AdamW do not assume that we want to stay close to our model's start state, and thus lead to models that tend to compromise performance on old data when fine-tuning on new data, also known as catastrophic forgetting McCloskey and Cohen [1989]. Our method serves as a regularization approach to bridge the gap in existing optimizers to mitigate this forgetting issue.

**Continual Learning**    Various approaches have been developed Kirkpatrick et al. [2017], Chaudhry et al. [2020], Jung et al. [2020], Titsias et al. [2020], Mirzadeh et al. [2021], in which the goal is to keep the information learned from the past tasks during continual learning. Learning Without Forgetting (LWF) Li and Hoiem [2017] stores the response of the old model on new tasks/data and supervises the new model on these responses using distillation to prevent catastrophic forgetting Masana et al. [2022]. Our work draws inspiration from LWF's key idea of storing a pre-trained model's response,

---

**Algorithm 1** PROFIT: A fine-tuning optimizer

---

**Require:** Converged model $\mathcal{M}(\mathbf{x}; \theta)$ with trainable weights $\theta$ and input $\mathbf{x}$, to be trained on data from similar domain $\mathbf{X}'$ with loss $L$.
**Require:** Initialize reference model weights $\theta_{\text{ref}}$.
**Require:** Initialize batch size $B$, reference steps $n_{\text{ref}}$, training steps $n_{\text{steps}}$, standard optimizer $\mathbf{O}$ with learning rate $\lambda_{\text{main}}$, and reference optimizer $\mathbf{O}^{(\text{ref})}$ with learning rate $\lambda_{\text{ref}}$. Each optimizer takes as arguments the current weights and a gradient update direction, producing updated weight values.

1: **for** $n_{\text{step}}$ steps **do**:
2:      $\theta_{\text{ref}} \leftarrow \theta$                                          ▷ Save the model state.
3:      **for** $n_{\text{ref}}$ steps **do**
4:          Take new $B$ examples from $\mathbf{X}'$ and calculate gradients $\mathbf{g} := \nabla_\theta L$.
5:          Take one step with reference optimizer $\theta \leftarrow \mathbf{O}^{(\text{ref})}(\theta, \mathbf{g})$.
6:      **end for**

7:      Calculate $\Delta = \theta - \theta_{\text{ref}}$.          ▷ Calculate total displacement during reference steps.
8:      Find $\mathbf{g} := \nabla_\theta L$ for a new batch as in Line 4.
9:      Calculate dot product $\omega = \langle \Delta, \mathbf{g} \rangle$.
10:      **if** $\omega < 0$ **then**:
11:          $\mathbf{g} \leftarrow \mathbf{g} \perp \Delta$          ▷ $\mathbf{a} \perp \mathbf{b}$ denotes orthogonalizing $\mathbf{a}$ with respect to $\mathbf{b}$
12:      **end if**
13:      $\theta \leftarrow \theta_{\text{ref}}$.                                        ▷ Restore original state.
14:      $\theta \leftarrow \mathbf{O}(\theta, \mathbf{g})$                          ▷ Take step with main optimizer.
15: **end for**

---

but we aim to avoid the storage of old data/checkpoint/statistics. Instead, we use the network's response at the initial state of each iteration. Other directions for addressing catastrophic forgetting are rehearsal-based methods Rebuffi et al. [2017b], Chaudhry et al. [2019a], Lopez-Paz and Ranzato [2017], Chaudhry et al. [2019b], Saha et al. [2022] that directly make use of the old data source, and architecture-based methods that minimize inter-task interference through new architectures Mallya and Lazebnik [2018], Serrà et al. [2018], Li et al. [2019], Wortsman et al. [2020], Wu et al. [2019]. These methods add substantial infrastructural overhead, while our approach is attractive in its simplicity. Since PROFIT does not require replay buffer or architectural changes, we primarily benchmark against optimizers and fine-tuning techniques that practitioners would otherwise use.

## 3 Method

### 3.1 The PROFIT Optimizer

Implementing PROFIT (Algorithm 1) involves defining an optimizer wrapper that takes two standard optimizers $\mathbf{O}$ and $\mathbf{O}^{(\text{ref})}$ as inputs. The latter "reference" optimizer perturbs the system from equilibrium, while the former "main" optimizer uses this perturbation to make the final update. The entire logic of PROFIT can be encapsulated within the logic of this optimizer wrapper class, making the method very portable, modular, and invariant to the choice of "main" optimizer.

Given a model $\mathcal{M}$ with weights $\theta$ trained on data $\mathbf{X}$ corresponding to an old task, and a data source $\mathbf{X}'$ from a similar domain corresponding to a new task (with a task loss $L$), our objective is to fine-tune $\mathcal{M}$ to work well on both old and new tasks. We make a strict assumption about having a converged model $M$ available as input to PROFIT; untrained weights will lead to poor performance. Furthermore, our approach also requires two optimizers, a standard optimizer $\mathbf{O}$ and a reference optimizer $\mathbf{O}^{(\text{ref})}$. The user can tune $\mathbf{O}$ as per their needs, while we recommend using SGD for $\mathbf{O}^{(\text{ref})}$.

A step through our optimizer works as follows. First, we store the current state of $\mathcal{M}$ by saving $\theta$ in $\theta_{\text{ref}}$. Next, we draw $n_{\text{ref}}$ batches from $\mathbf{X}'$, one at a time, and iteratively minimize $L$ with the reference optimizer $\mathbf{O}^{(\text{ref})}$. We have now perturbed the system from equilibrium and must decide the best way to restore the said equilibrium.

To do so, we first calculate $\Delta := \theta - \theta_{\text{ref}}$, the displacement vector after $n_{\text{ref}}$ steps of the optimizer $\mathbf{O}^{(\text{ref})}$. We reason that if the stored equilibrium state corresponds to a good critical point of the original model, then $\Delta$ corresponds to the benign gradient direction that will restore the original critical point, as gradient descent would send us in the $-\Delta := \theta_{\text{ref}} - \theta$ direction. We thus have two potentially conflicting gradient updates: the one corresponding to $\Delta$, and the one corresponding to the next queried update by $\mathbf{O}^{(\text{ref})}$, which we call $\mathbf{g}$. We need to decide on how to take a single gradient step that is consistent with both options.

We then borrow an idea from PCGrad Yu et al. [2020], which reconciled conflicting gradients by orthogonally projecting them onto each other in a pairwise fashion. Crucially, we choose to project only $\mathbf{g}$ onto $\Delta$, and not the other way around, because $\Delta$ represents a gradient towards the old dataset that may no longer be accessible and therefore must be treated with more care. We end with the two gradient updates $\mathbf{g}$ and $\mathbf{g} \perp \Delta$, and take both steps by first restoring $\theta \mapsto \theta_{\text{ref}}$ and then allowing $\mathbf{O}$ to take a step in the $\mathbf{g} \perp \Delta$ direction. This process is repeated until training is complete.

This formulation allows us to view fine-tuning as temporal multi-task learning, with the two tasks being: (1) pretraining ($\Delta$), and (2) fine-tuning ($\mathbf{g}$). To the best of our knowledge, this is the first time it has been viewed through this lens, and this insight may pave avenues for future research.

## 3.2 Theoretical Considerations

We present key theoretical properties of PROFIT with proof sketches. First, we show that PROFIT is "correct" by decreasing the loss on the old task/setting. We also discuss potential failure cases by identifying all stable points of PROFIT, arguing that these are not problematic in practice.

**Theorem 3.1.** *(**Correctness on old data**) Take a model $\mathcal{M}(\mathbf{x}; \theta)$ converged on data $\mathcal{X}_{old}$ with loss $L_{old}$, which we would now like to fine-tune on data $\mathcal{X}_{new}$ with loss $L_{new}$. Suppose that $\mathbf{O}^{(ref)}$ takes $\theta \mapsto \theta'$. A single step of PROFIT in batch $\mathbf{x}_{new}$ with sufficiently low learning rates $\lambda_{main}, \lambda_{ref}$ will decrease $L_{old}(\mathcal{X}_{old})$ from its value at $\theta'$.*

*Proof.* At a convergence point local minimum $\mathbf{x}_0$, the first-order gradient vanishes, and to leading order the loss surface will look like $L(\mathbf{x}) \approx L(\mathbf{x}_0) + 0.5(\mathbf{x} - \mathbf{x}_0)^t \mathbf{H}_0(\mathbf{x} - \mathbf{x}_0)$ for positive definite hessian $\mathbf{H}_0$. The gradient within this region is therefore $\nabla L(\mathbf{x}) \approx \mathbf{H}_0(\mathbf{x} - \mathbf{x}_0) \equiv \mathbf{H}_0(\Delta)$. Because $\mathbf{H}_0$ is positive definite, we conclude that $(-\Delta)^t \mathbf{H}_0(\Delta) < 0$, which means that moving the system in the $-\Delta$ direction is a valid gradient descent direction. $\square$

The prior theorem establishes that PROFIT accomplishes precisely what it seeks to do: even if the old data are no longer available, PROFIT allows us to train as if we can still compute the full loss function of the old data. As the system moves further away from the old equilibrium, we are able to restore some of the function of that equilibrium through these regularized updates. In particular, there is a case where PROFIT leads to a trivial update. Intuitively, since the model converged, moving away from the minimum creates a gradient $(-\Delta)$ that points back toward it. PROFIT leverages this implicit gradient to regularize the update on the new task.

**Theorem 3.2.** *(**Stable points**) Suppose a model has weights $\theta$ and $\mathbf{O}^{(ref)}$ maps $\theta \mapsto \theta'$ and $\mathbf{O}^{(ref)}$ is SGD. If we are not at a critical point of $L_{new}$, PROFIT will result in zero change in the model weights $\theta$ if and only if $\hat{\nabla}_\theta L_{new} = \hat{\nabla}_{\theta'} L_{new}$, where $\hat{\nabla}$ refers to the unit vector corresponding to $\nabla$.*

*Proof.* In this case $\Delta$ would point in the direction $-\nabla_\theta L_{\text{new}}$, at which point $\nabla_\theta L_{\text{new}} \perp \Delta = 0$, and the total update from PROFIT will be just a restoration to $\theta$. If the equality of the gradient does not hold, $\nabla_\theta L_{\text{new}} \perp \Delta \neq 0$ will lead to a non-zero total update. $\square$

**Corollary 3.3.** *(**Linearity forces a stable point**) In the situation defined by Theorem 3.2, PROFIT will encounter a stable point if the loss surface is perfectly linear between $\theta$ and $\theta'$.*

The prior theorem and corollary demonstrate that PROFIT will fail to move the system when the loss surface becomes exactly locally linear at a point. This almost never occurs for high-dimensional loss surfaces that exist for deep models.

**Theorem 3.4.** *(**Convergence**) For a model trained with SGD $\mathbf{O}^{(ref)}$, any standard optimizer for $\mathbf{O}$, and loss $L$ under PROFIT, the model is guaranteed to converge to either (1) a stable point of the system as defined by Theorem 3.2, or (2) a convergence point of $\mathbf{O}$.*

*Proof.* If the system does not converge to a stable point as defined in Theorem 3.2, then it will necessarily produce a valid gradient descent direction because it is an orthogonal projection of a valid gradient direction $\nabla_\theta L$. As such, it inherits the same stable points as $\mathbf{O}$. $\qquad\square$

In summary, PROFIT enjoys the following theoretical guarantees:

1. By Theorem 3.1, it mitigates catastrophic forgetting relative to prior methods.
2. By Theorem 3.2, it introduces only rare failure modes, which require all higher-order gradients to vanish.
3. By Theorem 3.4, it inherits the stable points of its parent optimizers, allowing the use of state-of-the-art and future optimizers.

### 3.3 Hyperparameter Discussion

PROFIT introduces three main hyperparameters for fine-tuning: $n_{\text{ref}}$, $\mathbf{O}^{(\text{ref})}$, and $\lambda_{\text{ref}}$. $n_{\text{ref}}$ controls the degree of exploration of the reference optimizer, while $\lambda_{\text{ref}}$ controls the step-size at each iteration of the reference step. The choice of $\mathbf{O}$ can be set by the practitioner according to their needs, but $\mathbf{O}^{(ref)}$ should be set to standard SGD, as the gradient calculations that drive PROFIT are the cleanest when the reference updates are simple.

A primary concern is the cost of setting $n_{\text{ref}}$, as it requires $n_{\text{ref}}$ additional optimization steps per training step. In practice, the practitioner is encouraged to start with $n_{\text{ref}} = 1$, which works well in practice, and to only increase $n_{\text{ref}}$ if performance is lacking. However, we note two reasons why the potentially increased compute of PROFIT is not a large issue: (1) we find that PROFIT generally converges faster, possibly due to the positive regularization effects of the method, and (2) fine-tuning is generally run for much fewer steps than training from scratch, so the effects of this additional training time are often still very tractable. For all experiments, we use $n_{\text{ref}} = 1$.

Mathematically, $\lambda_{\text{ref}}$ encodes how quickly we travel from equilibrium to study the general shape of the loss surface, so that we can make an informed decision on where to go next. In practice, we find that there is often a sweet spot somewhere between $\lambda_{main}/10$ and $\lambda_{main}/10000$, but the exact value is highly dependent on the loss surface for the specific problem. In fact, it may turn out that the optimal value of $\lambda_{\text{ref}}$ tells us something about the fundamental properties of the loss surface of that particular setting, but such analysis falls outside the scope of the present work.

### 3.4 Memory Considerations

One of the primary difficulties in implementing PROFIT is that storing the reference weights $\theta_{\text{ref}}$ as defined in Algorithm 1 will increase the memory footprint. In memory profiling experiments (Appendix D.1), we found that a vanilla implementation of PROFIT will on average increase the memory overhead of model training by approximately 25%, which then decreases to 0% if we remove the additional memory consumed by $\theta_{\text{ref}}$. In many fine-tuning scenarios, this additional memory consumption may not pose a problem, as model backbones may be largely frozen during fine-tuning. However, it is still worthwhile to address this substantial memory footprint.

In general, modern optimizers such as AdamW Loshchilov and Hutter [2019] keep reference statistics of model weights in memory which also incurs substantial memory penalties. It may be possible to implement memory-efficient approximate versions of PROFIT for most popular deep learning optimizers by using the already-allocated extra memory within those optimizers to also perform the PROFIT computations. For example, such an efficient implementation for momentum might involve replacing $\theta_{\text{ref}}$ with $\theta - k\mathbf{m}$ for momentum vector $\mathbf{m}$ and some appropriate normalization constant $k \sim \lambda||\theta - \theta_{\text{ref}}||/||\mathbf{m}||$. This implementation would reduce the memory footprint of PROFIT to a single additional scalar ($||\theta - \theta_{\text{ref}}||$). We treat these memory-efficient implementations of PROFIT as high-priority directions for future work.

### 3.5 Assumptions

The key assumptions made by PROFIT are:

| Method | Original Data Error ($\downarrow$) | New Data Error ($\downarrow$) |
|---|---|---|
| Baseline Trained on Original Data | 0.0054 | 1.907 |
| Fine-Tune on New Data (Full Model) | 0.705 | 0.504 |
| Fine-Tune on New Data (Head Only) | 0.110 | 0.572 |
| PROFIT on New Data | **0.046** | **0.501** |

Table 1: Results on a 2D toy problem. A baseline is trained on just the original data domain and then fine-tuned at a lower learning rate on a new data domain (both the full model and the head only). This is compared to PROFIT, which is trained on the full model. Fine-tuning on the model head weights only provides some protection against performance regression on the original data, but also is less able to adapt to the new data. PROFIT outperforms both baselines and shows impressive resilience in maintaining performance on the original split. All results have standard error $\leq 0.01$.

1. Proximal Fine-tuning: pre-training and fine-tuning datasets come from similar distributions or modalities.

2. Well-trained: the initial network is converged and not randomly initialized.

Orthogonalizing between **g** and $\Delta$ implies that we are interested in resolving the conflict between the original and fine-tuning datasets. If the original and fine-tuning distributions are significantly different, the orthogonalized gradient direction may lead to destructive interference. Therefore, our interpretation holds only if the original and fine-tuning datasets are from proximal distributions. Assumption 1 may differ from typical fine-tuning literature and appear greatly limiting, but is exceedingly common in practice. For example, a self-driving car may train a classification head for a new scooter type while maintaining performance on existing vehicles. Assumption 2 is required since our method is exclusively designed for fine-tuning (which is a guiding principle). Nonetheless, we include it here to emphasize to the reader that PROFIT cannot be used for general model optimization.

## 4 Experiments

We now detail a number of experiments for PROFIT in diverse settings: image classification, visual task adaptation, and large-scale motion prediction. We primarily focus on comparisons to standard fine-tuning with commonly used optimizers (on either the full model or just the model head), as those are - by a large margin - still the most commonly used fine-tuning methods in the industry due to their known performance and ease of implementation. We will show that PROFIT, while easy to implement, provides a significant performance boost in all cases.

### 4.1 A Simple Toy Example

We first apply PROFIT to a simple, 2D regression problem with MLPs as a toy example. We choose the 2D function $f(\mathbf{x}) = \sin(10|\mathbf{x}|)$, as radially symmetric and periodic functions pose some challenge for neural networks models. We also add $\mathcal{N}(0, 1)$ noise to the output. Even though we want to fit to a low-dimensional example, it is still important for the problem to be difficult enough to see interesting behavior within our models, especially given the dimensionality requirements as discussed in Section 3.2 of the main paper. The original dataset consists of input-output pairs where the input coordinates are drawn independently from $\mathcal{U}[-1.0, 1.0]$, while the new dataset we wish to fine-tune on has input coordinates drawn from $\mathcal{U}[0.8, 1.5]$. This is clearly also challenging because, although the domains of the two datasets overlap in the interval $[0.8, 1.0]$, they are largely non-overlapping.

Our MLP consists of three layers with weights of shape $[2, 500] \rightarrow [500, 500] \rightarrow [500, 1]$, and we use RMSProp as our baseline optimizer. The baseline model is trained on the original data split only for 10000 steps at a learning rate of $1e-2$, with fine-tuning runs trained at a learning rate of $5e-4$ for 1500 steps. For exact details on the training procedure, please refer to the Appendix C.1.

The results are visualized in the bottom half of Figure 1, with more granular visualizations in Appendix C.1. Visually, the benefits of PROFIT are pronounced; standard fine-tuning creates a warped shape bearing minimal resemblance to the original ground truth. PROFIT effectively remembers the original shape with only minimal regressions along the steep edges of the distribution.

| Method | SGD | Adam | Lookahead | Adan | PROFIT |
|---|---|---|---|---|---|
| ResNet-18 | 24.49 / 74.59 | 34.64 / 73.11 | 26.10 / 73.82 | 35.17 / 72.70 | **35.26 / 74.70** |
| ViT-Tiny | 53.63 / 61.98 | 56.00 / 58.99 | 55.64 / 61.35 | 53.04 / 61.62 | **58.53 / 62.20** |
| ViT-Small | 57.93 / 65.04 | 58.60 / 63.93 | 57.81 / 65.27 | 53.85 / 64.09 | **59.02 / 65.44** |

Table 2: Image classification accuracy (%): CIFAR10 / CIFAR100 after fine-tuning on CIFAR100. Each cell shows (CIFAR10 / CIFAR100). PROFIT achieves the best balance between transfer and forgetting. Compared methods include Adam (Kingma and Ba [2015]), Lookahead (Zhang et al. [2019]), and Adan (Xie et al. [2024]).

| | Natural | | | | | | | Specialized | | | | Structured | | | | | | | |
|---|---|---|---|---|---|---|---|---|---|---|---|---|---|---|---|---|---|---|---|
| | Cifar100 | Caltech101 | DTD | Flower102 | Pets | SVHN | Sun397 | Camelyon | EuroSAT | Resisc45 | Retinopathy | Clevr-Count | Clevr-Dist | DMLab | KITTI-Dist | dSpr-Loc | dSpr-Ori | sNORB-Azim | sNORB-Ele |
| Full Jia et al. [2022] | 68.9 | 87.7 | 64.3 | 87.2 | 86.9 | 87.4 | 38.8 | 79.7 | 95.7 | 84.2 | 73.9 | 56.3 | 58.6 | 41.7 | 65.5 | 57.5 | 46.7 | 25.7 | 29.1 |
| Linear Jia et al. [2022] | 64.4 | 85.0 | 63.2 | 97.0 | 86.3 | 36.6 | 51.0 | 78.5 | 87.5 | 68.5 | 74.0 | 34.3 | 30.6 | 33.2 | 55.4 | 12.5 | 20.0 | 9.6 | 19.2 |
| VPT Jia et al. [2022] | **78.8** | **90.8** | 65.8 | **98.0** | 88.3 | 78.1 | 49.6 | 81.8 | **96.1** | 83.4 | 68.4 | **68.5** | **60.0** | 46.5 | 72.8 | 73.6 | 47.9 | **32.9** | **37.8** |
| PROFIT | 58.9 | 55.6 | 51.4 | 92.4 | 50.8 | 19.6 | 37.7 | 79.3 | 53.8 | 43.5 | 73.5 | 12.6 | 37.9 | 21.1 | 77.5 | 10.2 | 24.6 | 23.6 | 10.7 |
| PROFIT (warm-up) | 69.4 | 90.5 | **69.6** | **98.9** | **89.2** | **89.4** | **52.0** | **84.9** | **95.7** | **85.6** | **74.3** | **80.5** | **64.7** | **51.9** | **80.6** | **82.3** | **49.3** | 32.7 | 35.1 |

Table 3: Image classification accuracy on VTAB-1K. **Bold** = best, Underline = second-best. Ties for best are all **bolded**; no second-best is shown in such cases. We compare PROFIT against standard fine-tuning strategies, including full fine-tuning and linear fine-tuning (denoted by Full and Linear). A naive PROFIT violates the proximity condition and performs poorly, but with AdamW warm-up it achieves consistently higher accuracy.

Numerically, as shown in Table 1, after fine-tuning with different techniques, it is clear that PROFIT outperforms the baseline in not forgetting the original dataset distribution. Even though PROFIT modifies all model weights, it mitigates forgetting relative to the head-only fine-tuning baseline by a sizable margin, even though PROFIT acts on significantly more weights. Fine-tuning of the full model leads to disastrous results, with significant deformations of the predictions. PROFIT allows us the flexibility of full-model fine-tuning without the drawbacks of catastrophic forgetting.

## 4.2 Image Classification

Next, we demonstrate the effectiveness of PROFIT for image classification. CIFAR10 and CIFAR100 (Krizhevsky and Hinton [2009]) are the de facto benchmarks for image classification. The dataset consists of $32 \times 32$ images and the task is to classify an image into one of $K$ categories.

We first train a network on CIFAR10 (pre-training) and fine-tune the network on CIFAR100. As discussed, PROFIT assumes proximality, i.e., both the pre-training and fine-tuning datasets need to be from a similar domain distribution. This assumption is met for our choice of CIFAR10 and CIFAR100, as they are both labeled subsets of Tiny Images Torralba et al. [2008].

We experiment with various backbones, including ResNet-18 (He et al. [2016]), ViT-Tiny, and ViT-Small(Dosovitskiy et al. [2021]), and compare PROFIT against popular optimizers like SGD, Adam (Kingma and Ba [2015]), Lookahead (Zhang et al. [2019]), and Adan (Xie et al. [2024]).

Table 2 shows that PROFIT outperforms standard fine-tuning across all backbones and optimizers. CIFAR10 accuracy after fine-tuning is also higher, indicating better retention of the original task. For example, with ViT-Tiny, Adam fine-tuning results in 55.64% (CIFAR10) and 61.35% (CIFAR100), compared to PROFIT's 58.53% and 62.20%. Similarly, fine-tuning ViT-Small using Adam yields 58.60% (CIFAR10) and 63.93% (CIFAR100), while PROFIT achieves 59.02% and 65.44%.

## 4.3 Visual Task Adaptation Benchmark

The VTAB-1K Zhai et al. [2019] dataset is a popular representation learning benchmark that evaluates generalization in 19 diverse classification tasks. It aims to learn feature representations effective for all tasks, with performance measured by classification accuracy on each.

We use ViT-Base Jia et al. [2022] with ImageNet pretraining as the backbone for our experiments. This setting is an example of a setting in which the original (ImageNet) and fine-tuning distributions

| Backbone | Optimizer | Accuracy (%) ↑ | GPT Score (%) ↑ | Match Score (%) ↑ | BLEU-4 ↑ | ROUGE-L ↑ | CIDEr ↑ | Final score ↑ |
|---|---|---|---|---|---|---|---|---|
| LLaMA-Adapter-v2 Gao et al. [2023] | AdamW | 62.21 | 70.57 | 36.33 | 0.541 | 7.16 | 0.023 | 56.98 |
| | PROFIT | 67.88 | 72.12 | 37.37 | 0.557 | 7.28 | 0.035 | 59.16 |

Table 4: Results on the DriveLM Benchmark. We fine-tune LLaMA-Adapter-v2 Gao et al. [2023] and compare PROFIT with AdamW, the de-facto method for fine-tuning VLMs. Our method shows improvements on all metrics, showcasing the applicability of PROFIT to large foundational models.

(VTAB-1K) differ significantly, violating Assumption 1 (Sec. 3.5). Although our method is not well tuned for these fine-tuning settings, this situation is common in practice.

First, we show our results in Table 3, and compare our method with AdamW Loshchilov and Hutter [2019] fine-tuning (Full – row 1) as well as final layer fine-tuning (Linear – row 2). Full fine-tuning using PROFIT does not work well, showing poor performance across all 19 tasks in the benchmark. For example, PROFIT achieves 12.6% accuracy on the Clevr-Count benchmark while Visual Prompt Tuning Jia et al. [2022] achieves 68.5% accuracy.

However, we provide a training recipe on how PROFIT can be used in such settings. First, we fine-tune the model towards the target distribution with an AdamW warmup for 10 epochs, and then apply PROFIT to fine-tune for the remaining 90 more epochs (PROFIT (warmup)). Intuitively, this moves the model's distribution somewhere between the pre-training and fine-tuning distribution, which makes it amenable to the structure of PROFIT. This approach outperforms the full model fine-tuning with AdamW. So, while our method as expected is not recommended for non-proximal settings (as discussed in Section 3.5), using a short warm-up phase with another optimizer allows PROFIT to substantially outperform full fine-tuning with standard optimizers.

## 4.4 Multimodal Vision-Language Models (VLM)

VLMs (Hwang et al. [2025]) trained on web-scale data have helped to improve generalization of end-to-end driving systems. DriveLM (Sima et al. [2023]) is a visual question-answering (VQA) benchmark to evaluate VLMs on autonomous driving. The DriveLM-nuScenes dataset contains sensor inputs and a text prompt, and the network aims to provide accurate textual responses for prompts which probe perception, prediction, and planning capabilities respectively.

We use a pre-trained LLaMA-Adapter-v2 (Gao et al. [2023]), which performs bias tuning on LLaMA-7B (Touvron et al. [2023b]). We compare PROFIT with AdamW (Loshchilov and Hutter [2019]), the de facto method for fine-tuning VLMs. We report results on the validation set in Table 4. PROFIT improves accuracy over AdamW by 5.6%, GPT score by 1.5%, Match Score by 1%, and the Final Score by 2%. In addition, PROFIT provides better results for VQA, as seen by improved performance metrics (BLEU-4, ROUGE-L, CIDEr). For further explanation on metrics, we refer the reader to the Sec. C.4 of the appendix.

## 4.5 Large-Scale Robotics Motion Prediction

We evaluate PROFIT on the Waymo Open Motion Dataset (WOMD) Ettinger et al. [2021], a large-scale driving dataset. The task is to predict agent trajectories over 8 seconds using multi-modal observations from the last second, including agent histories, map data, and traffic light states. Our model builds on the state-of-the-art Wayformer Nayakanti et al. [2022] by fusing multi-modal inputs with a self-attention transformer and predicting future trajectories using learned latent queries.

We first train a model on car trajectory prediction, and then fine-tune the model on data from the same class (car) as well as different classes (pedestrian). For our baselines, we use the AdamW Loshchilov and Hutter [2019] optimizer. WOMD allows us to evaluate PROFIT on (1) fine-tuning on the same data and (2) on different domain-shifted tasks. For example, CIFAR100 is similar to CIFAR10, whereas pedestrian trajectories differ from car trajectories.

Figure 2 and Table 5 show the results. Average Distance Error (ADE) measures the mean distance between ground truth and predictions at each point, while Final Distance Error (FDE) assesses only the final point. PROFIT consistently outperforms the baseline, especially in car-to-pedestrian fine-tuning. Fine-tuning the head alone performs poorly on car-to-pedestrian tasks, and full fine-tuning is better, but still inferior to PROFIT. Thus, PROFIT effectively repurposes models for related settings.

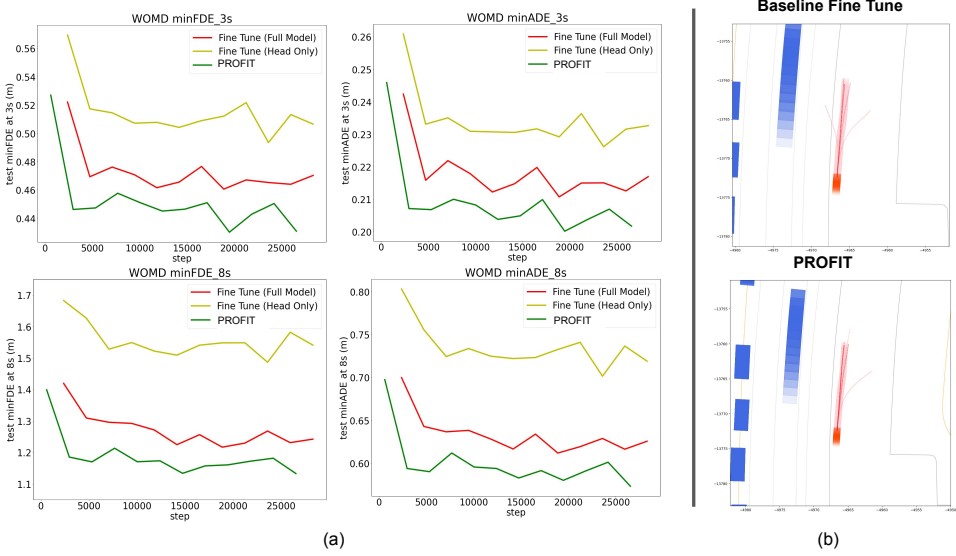

Figure 2: Results for PROFIT on Waymo Open Motion Dataset. (a) Training error curves for FDE at 3s and 8s (top) and ADE at 3s and 8s (bottom). PROFIT outperforms both fine-tuning baselines by a sizable margin. (b) Visualizations of motion prediction outputs for both the baseline fine-tune model (top) and PROFIT (bottom). Trajectory ground truth is shown as a shaded bar and denser lines represent more confident predictions. PROFIT (bottom) produces more confident predictions that align better with the ground truth (shaded bar) compared to the baseline (top). Best viewed in color.

| Method | Target Class | ADE@3s (m) | ADE@8s (m) | FDE@3s | FDE@8s |
|---|---|---|---|---|---|
| Baseline | - | 0.461 | 1.327 | 1.024 | 2.581 |
| fine-tune (F) | Car | 0.458 | 1.322 | 1.021 | 2.548 |
| fine-tune (H) | Car | 0.456 | 1.303 | 1.009 | 2.507 |
| PROFIT | Car | **0.454** | **1.299** | **1.008** | **2.489** |
| fine-tune (F) | Ped | 0.214 | 0.621 | 0.465 | 1.242 |
| fine-tune (H) | Ped | 0.232 | 0.724 | 0.508 | 1.544 |
| PROFIT | Ped | **0.203** | **0.579** | **0.427** | **1.145** |

Table 5: Results on Waymo Open Motion Dataset. F stands for full-model fine-tuning and H stands for head (last layer) only. Standard errors are within 0.005m for 3s metrics and 0.015m for 8s metrics. There is a minor but noticeable improvement for PROFIT on the car-to-car benchmarks and a substantial improvement for PROFIT on the car-to-ped benchmarks.

We also see minor improvements in the car-to-car benchmark, suggesting that PROFIT can be used to extract more performance from any model that has already converged. We could even imagine a scenario in which additional training using PROFIT is a standard model maintenance practice.

## 5   Conclusion

We introduce PROFIT, an optimizer that improves robustness against catastrophic forgetting during fine-tuning by using confidence in the model's prior converged state as a regularizer. PROFIT excels in various settings: (1) fine-tuning to CIFAR100 from CIFAR10, (2) fine-tuning on a new distribution in VTAB-1K by providing a train recipe, (3) fine-tuning large Vision-Language Models for autonomous driving and question-answering, and (4) fine-tuning on both new and identical tasks in large-scale motion prediction. In all cases, PROFIT outperformed standard fine-tuning methods. Importantly, the modularity of PROFIT as an optimizer allows it to integrate easily into training pipelines. We believe PROFIT is a valuable tool for practitioners and encourages the development of new optimizers that support fine-tuning as the primary deep learning paradigm.

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

# A   Appendix: Introduction

In the appendix, we provide the following:

- A theoretical discussion of PROFIT (Sec. B).
- Detailed setup, hyper-parameters, and ablations (Sec. C).
- Limitations of PROFIT (Sec. D).

# B   Additional Theoretical Discussion

Here, we provide a bit more exposition on the theoretical properties of PROFIT. In the main text (Section 3), we proposed two properties of PROFIT: (1) that PROFIT updates will reduce the loss value of the old loss on the old data, despite making no assumptions on access to the old data, and (2) that PROFIT has stable points on linear loss surfaces.

The implications of statement (1) are fairly straightforward, as it implies that we can train with the settings of the old system even when that old system falls out of scope. This is the primary feature of PROFIT and is the main proof that PROFIT is a meaningful regularization method. But the implications of (2) are more interesting. The fact that PROFIT works despite not functioning properly on linear loss surfaces implies that PROFIT relies on nontrivial values of second-order gradients and curvature within the loss surface to function. Thus, any critical point to which the system converges while under PROFIT updates must have been reached through a nonlinear path from the model's starting point. This requirement may have robustness implications for the convergence points found by PROFIT, as any such convergence points must have alternative nonlinear paths leading to it.

However, there is an important discussion to be had regarding the convergence of PROFIT. We did not expand on this convergence in the main text because talking about convergence within a fine-tuning setting is potentially ambiguous, as it is difficult to deconvolve convergence on the new dataset with convergence on the old setting (for which we may no longer have available data). Especially when the number of incremental training stages increases, it becomes secondary to converge on the new (potentially small) dataset on which the model is fit, and more important to maintain the efficacy of the base model. In that case, the correctness property (1) becomes the main property of importance.

In general, a proof of convergence of PROFIT is difficult for two reasons: (1) The orthogonalization procedure we use is asymmetric versus the symmetric procedure in Yu et al. [2020], due to our prioritization of mitigating regressions, and (2) the restoration step to the old state after steps of $\mathbf{O}^{(\text{ref})}$ is discrete rather than treated as a separate incremental gradient step. Both of these design decisions were made to support the main goal of regression mitigation for applications of PROFIT. So, although we do not provide a complete proof of conditions under which PROFIT converges in this work, we make this omission precisely because convergence on the new data is a secondary goal in our setting, and the majority of our important design decisions were not made in service to that particular goal, but rather to the more difficult goal of regularization during fine-tuning.

# C   Experiments

## C.1   A Simple Toy Example

We provide more granular visualizations of toy example results in Figure 3. We note that even fine-tuning on the head layer only produces sizable shifts in the overall height of the output shape. while PROFIT more effectively remembers the original shape.

Our toy example ground truth is the function $f(\mathbf{x}) = \sin(10|\mathbf{x}|)$ with input in $\mathbb{R}^2$. This function was chosen because of its extreme nonlinearity and difficulty in fitting by standard neural networks. To further increase the challenge, for the training data, normal noise of size $\mathcal{N}(0,1)$ is added, while no noise is added to the test data. The "original dataset" consists of 50000 points with both dimensions between -1 and 1, while the "fine-tune dataset" consists of 50000 points with both dimensions between 0.8 and 1.5.

The model itself is a 3-layer MLP, consisting of weight layers [2, 500], [500, 500], and [500, 1]. LayerNorm (Ba et al. [2016]) is applied after every layer except for the last. RMSProp is used with

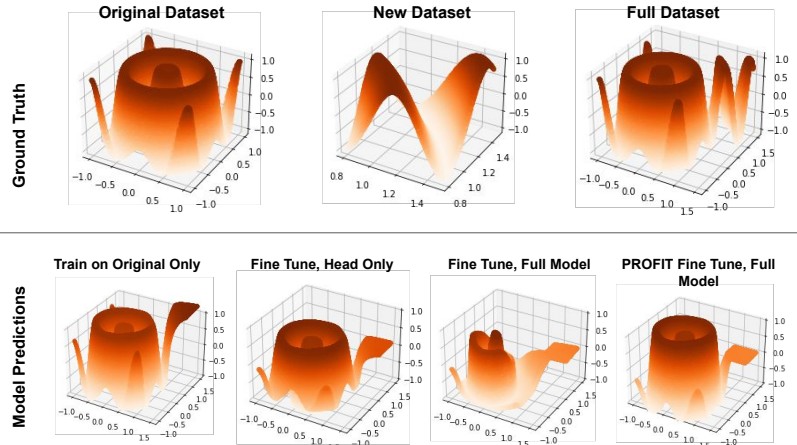

Figure 3: Toy example visualizations. The top row shows ground truth distributions for the original, new, and combined datasets, while the bottom row depicts model predictions for different training strategies: training on original data only, head-only fine-tuning, full model fine-tuning, and using the proposed PROFIT optimizer for full model fine-tuning. This highlights the PROFIT optimizer's effectiveness in retaining old task knowledge while adapting to new tasks. Best viewed in color.

| Backbone | CIFAR10 Accuracy |
|---|---|
| ResNet18 | 91.06% |
| ViT-Tiny | 82.38% |
| ViT-Small | 83.86% |

Table 6: Accuracy of pre-trained models on the CIFAR10 dataset (before fine-tuning).

default PyTorch hyperparameters ($\alpha = 0.99$, $\epsilon = 1e - 8$) and the learning rate $1e - 2$ for fitting to the original distribution, with a learning rate decay of 0.9 every 500 steps. After fitting the original distribution, we fine-tune to the new distribution for 1500 steps at a learning rate of $5e - 4$, with a decay factor of 0.95 every 100 steps. PROFIT is run with $n_{\mathrm{ref}} = 1$.

## C.2 Image Classification

### C.2.1 Pre-trained model performance

We show the performance of the pre-trained backbone on CIFAR10 datasets in Table 6. ResNet-18 achieves an accuracy of 91.06%, ViT-Tiny achieves 82.38% and ViT-Small achieves 83.86% accuracy.

### C.2.2 Comparison to Adapters

Adapters provide an efficient mechanism to fine-tune large pre-trained models with a fraction of trainable parameters. Although PROFIT does not introduce any additional parameters, we compare our method with commonly used parameter-efficient fine-tuning methods in Table 7.

Despite using fewer trainable parameters, our method marginally outperforms LoRA Hu et al. [2022] and VPT Jia et al. [2022] for ViT backbones and ConvAdapter Chen et al. [2024] for ResNet. Furthermore, our method is complementary to adapters and shows improvements when used as the optimizer for training adapter-based methods.

### C.2.3 Ablation Study

We ablate $n_{\mathrm{ref}}$ in Table 8. In general, we observe that increasing $n_{\mathrm{ref}}$ leads to better performance in the original task while compromising performance in the fine-tuning task. This follows from our discussion in Sec 3.2 of the main paper, since $n_{\mathrm{ref}}$ controls the degree of exploration away from the stable point. However, we recommend $n_{\mathrm{ref}} = 1$ as a starting point.

| Backbone | Adapter | Optimizer | CIFAR100 Accuracy |
|---|---|---|---|
| ResNet18 | – | PROFIT | 74.70% |
| | ConvAdapter (Chen et al. [2024]) | AdamW | 74.81% |
| | ConvAdapter (Chen et al. [2024]) | PROFIT | **75.26%** |
| ViT-Tiny | – | PROFIT | **62.20%** |
| | LORA (Hu et al. [2022]) | Adam | 61.02% |
| | LORA (Hu et al. [2022]) | PROFIT | 61.55% |
| | VPT (Jia et al. [2022]) | Adam | 60.63% |
| | VPT (Jia et al. [2022]) | PROFIT | 61.24% |
| ViT-Small | – | PROFIT | **65.44%** |
| | LORA (Hu et al. [2022]) | Adam | 64.25% |
| | LORA (Hu et al. [2022]) | PROFIT | 65.02% |
| | VPT (Jia et al. [2022]) | Adam | 63.98% |
| | VPT (Jia et al. [2022]) | PROFIT | 64.57% |

Table 7: CIFAR-100 accuracy for different backbones, adapters, and optimizers

| Method | $n_{ref}$ | CIFAR10 Acc ($\uparrow$) | CIFAR100 Acc ($\uparrow$) |
|---|---|---|---|
| ResNet-18 | 1 | 35.26% | **74.70%** |
| | 2 | 32.55% | 73.57% |
| | 5 | **39.27%** | 71.42% |
| ViT-Tiny | 1 | **56.75%** | **62.35%** |
| | 2 | 53.10% | 61.67% |
| | 5 | 56.56% | 59.14% |
| ViT-Small | 1 | **59.02%** | **65.44%** |
| | 2 | 58.58% | 64.91% |
| | 5 | 56.54% | 63.75% |

Table 8: Ablation study on $n_{ref}$ for image classification for different backbones. The results demonstrate that varying $n_{ref}$ can have a significant impact on model accuracy when using PROFIT.

We also ablate performance on $\frac{\lambda_{main}}{\lambda_{ref}}$ in Table 9, which shows that the best choice may vary for the choice of backbone. In general, larger values of $\lambda_{ref}$ promote new task accuracy, while smaller values effectively mitigate catastrophic forgetting. These results are reasonable, as smaller values of $\lambda_{ref}$ correspond to the reference point lying closer to the original model state. However, we were always able to beat the baseline on both old and new task accuracies simultaneously.

We also use this setting to substantiate Assumption 2 (Sec 3.4) which states PROFIT does not work well without a converged model. When we train a ResNet-18 from scratch on CIFAR100 using PROFIT, we get 1.05% accuracy, which is as good as random guessing.

### C.2.4 Regularization effects

We plot the training and validation losses for CIFAR100 fine-tuning in Figure 4. We observe similar trends for all network choices, where the validation losses illustrate that PROFIT is able to achieve better generalization across both the pre-training (CIFAR10) and fine-tuning (CIFAR100) tasks. Such results indicate that PROFIT may have positive regularization effects during the course of fine-tuning, allowing the network to converge quickly while achieving better performance.

### C.2.5 Implementation Details

For CIFAR10 pre-training, we use Adam optimizer with a learning rate of $1e-4$. ViT-Tiny and ViT-Small are trained for 400 epochs, while ResNet-18 is trained for 200 epochs. For the Lookahead optimizer, use $\alpha = 0.5$ and $k = 1$, for fair comparison against our method which uses $n_{ref} = 1$. We perform a parameter sweep to obtain the best performance for each method on CIFAR100 fine-tuning, and list the best obtained hyper-parameters in Table 10.

### C.2.6 Memory Footprint

As highlighted in Section 3.4 in the main paper, one of the main limitations of our work is the additional memory consumed. We use 4 Tesla T4 GPUs for all our experiments and quantify the increase in GPU memory consumption and train time by PROFIT in Table 11. Fine-tuning ResNet-18 with PROFIT consumes an additional 2.23 GB of GPU memory, while fine-tuning ViT-Tiny and

| Method | $\frac{\lambda_{\text{main}}}{\lambda_{\text{ref}}}$ | CIFAR10 Acc (↑) | CIFAR100 Acc (↑) |
|---|---|---|---|
| ResNet-18 | 10 | 35.26% | **74.70%** |
| | 100 | 35.16% | 74.45% |
| | 1000 | 34.35% | 74.27% |
| | 10000 | **35.83%** | 74.41% |
| ViT-Tiny | 10 | 57.32% | 61.99% |
| | 100 | 56.75% | **62.35%** |
| | 1000 | **58.53%** | 62.20% |
| | 10000 | 53.78% | 62.05% |
| ViT-Small | 10 | 58.41% | **66.29%** |
| | 100 | **59.02%** | 65.44% |
| | 1000 | 56.96% | 65.53% |
| | 10000 | 56.84% | 65.95% |

Table 9: Ablation study on $\frac{\lambda_{\text{main}}}{\lambda_{\text{ref}}}$ for PROFIT in image classification for different choices of backbone architectures. For ResNet-18, smaller values of $\frac{\lambda_{\text{main}}}{\lambda_{\text{ref}}}$ yield relatively stable accuracy, while ViT-based models show a more pronounced change, with peak performance observed at specific ratios. This result highlights the importance of fine-tuning $\frac{\lambda_{\text{main}}}{\lambda_{\text{ref}}}$ to achieve optimal performance for different backbone architectures in image classification tasks.

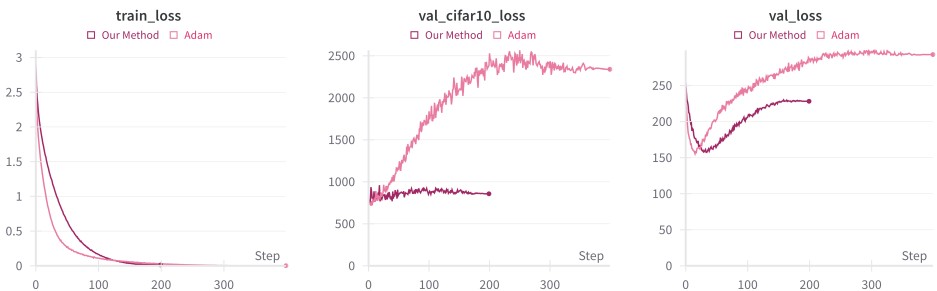

Figure 4: Training and validation losses for fine-tuning ViT-Small on CIFAR100.

ViT-Small consumes 1.3 GB and 1.1 GB extra. This increase is a consequence of loading the reference optimizer states in memory. However, our method does not introduce any new learnable parameters.

### C.3 Large-Scale Robotics Motion Prediction

The motion prediction model follows a standard encoder-decoder transformer architecture, as in Wayformer Nayakanti et al. [2022].

The encoder takes multi-modal inputs as the target agent's history, nearby agent histories, map information, and traffic light states. Each input modality is encoded by a separate MLP to an embedding with a dimension of 64. The input embeddings are fused through concatenation as input tokens to a self-attention transformer. The encoder transformer includes 2 attention layers, 8 heads, 256 hidden dimensions, and 1024 feedforward dimensions. We add learned positional embeddings, initialized as a Gaussian vector with zero mean and standard deviation of 0.02, to each token.

The decoder is a cross-attention transformer that attends six learnable latent queries, initialized with zero mean and standard deviation of 0.02, to encoder embeddings. The decoder transformer includes 8 attention layers, 8 heads, 256 hidden dimensions, and 1024 feedforward dimensions. The output queries are mapped to a weighted set of six trajectory samples through an MLP. Each sample includes $(x, y)$ positions for the next 80 timesteps and a weight scalar.

The model is trained end-to-end by a smooth L1 loss on the trajectory predictions and a cross-entropy loss on the predicted weights. The AdamW optimizer is used with default PyTorch hyper-parameters: learning rate $= 1e - 3$, $\beta s = (0.9, 0.999)$, weight decay $= 1e - 2$. The base model is trained on the WOMD training set for 60 epochs with a batch size of 256.

| Method | Optimizer | Learning Rate | Epochs |
|--------|-----------|---------------|--------|
| ResNet-18 | Adam | 1e-4 | 200 |
| | Lookahead | 1e-4 | 100 |
| | PROFIT | 1e-4 | 100 |
| ViT-Tiny | Adam | 1e-5 | 400 |
| | Lookahead | 1e-4 | 200 |
| | PROFIT | 1e-4 | 200 |
| ViT-Small | Adam | 3e-4 | 400 |
| | Lookahead | 3e-4 | 200 |
| | PROFIT | 3e-4 | 200 |

Table 10: Hyper-parameters for CIFAR100 results (Sec 4.1 in the main paper).

| Method | Optimizer | Time (sec / epoch) | GPU Memory (GB) |
|--------|-----------|--------------------|-----------------|
| ResNet-18 | Adam | 185 | 5.50 |
| | PROFIT | 248 | 7.73 |
| ViT-Tiny | Adam | 135 | 4.45 |
| | PROFIT | 185 | 5.86 |
| ViT-Small | Adam | 198 | 4.17 |
| | PROFIT | 269 | 5.28 |

Table 11: Training time and GPU Memory utilization of different methods for CIFAR100 fine-tuning.

For car-to-car fine-tuning experiments, learning rate is dropped by a factor of 100 from the original and training is performed for only 1500 steps (because the original training run already converged, training for too long in this setting leads to overfitting). For car-to-pedestrian fine-tuning experiments, learning rate also is dropped by a factor of 100, but training is allowed to run for the same number of steps as the original model.

For PROFIT, $n_{\text{ref}}$ is set to 3 and $\lambda_{\text{ref}}$ is set to 1/10 of the learning rate. We note that the value of $n_{\text{ref}}$ is quite high in this setting, but training is relatively fast and you can get better results fairly early on in training so the number of steps can be cut down considerably (see, for example, the curves in Figure 3 of the main paper).

## C.4 Multimodal Vision Language Models

### C.4.1 Ablation Study

The DriveLM benchmark leverages a variety of VQA metrics for evaluation. BLEU Papineni et al. [2002] evaluates precision by measuring n-gram similarity between generate and reference texts. ROUGE-L Lin [2004] evaluates recall by measuring the longest common subsequence between the generated and reference texts. CIDEr Vedantam et al. [2015] evaluates quality by computing the cosine similarity between the n-gram TF-IDF vectors for the generated and reference sentences. GPT Score aims to captures semantic nuances missed by aforementioned metrics by using ChatGPT (GPT-3.5-Turbo) as an evaluator. In addition to these metrics, perception accuracy is evaluated using the ground-truth objects in the scene, while prediction accuracy is evaluated over discretized future states. Match score evaluates whether the VLM correctly understands the order in which to attend to other agents in the scene. The final score for this benchmark is a weighted average of these metrics. For further details, we refer the reader to the evaluation criteria for the DriveLM Challenge contributors [2023].

We provide an inference visualization of our method in Figure 5. The baseline, fine-tuned with AdamW, is unable to detect the traffic light and black sedan in the scene. This leads to the suggested plan of running the red light, which is a potentially catastrophic error. On the other hand, PROFIT successfully identifies the red light and follows traffic regulations by planning to remain stationary. We conclude PROFIT is a useful tool in the era of VLMs to extract better performance on a new setting.

We ablate performance on $\frac{\lambda_{\text{main}}}{\lambda_{\text{ref}}}$, a key hyper-parameter to PROFIT, in Table 12. In general, larger values encourage exploration and improves fine-tuning task accuracy, as discussed in Sec 4.3 of the main paper.

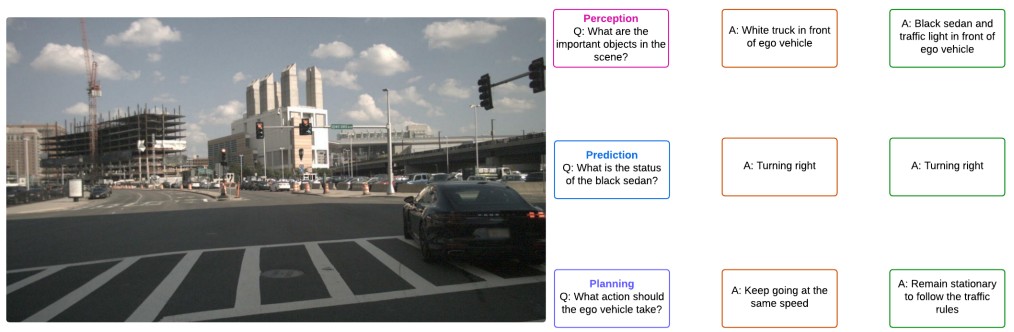

Figure 5: We compare the baseline with PROFIT on an example from DriveLM. The model fine-tuned with PROFIT is able to perceive the traffic light and black sedan in the scene, while the baseline (fine-tuned with AdamW) does not detect the traffic light and hallucinates the presence of a white truck. Consequently, the baseline suggests running the red light, while our method follows traffic rules by staying stationary. Best viewed in color.

| $\frac{\lambda_{\text{main}}}{\lambda_{\text{ref}}}$ | Accuracy (%) ↑ | GPT Score (%) ↑ | Match Score (%) ↑ | BLEU-4 ↑ | ROUGE-L ↑ | CIDEr ↑ | Final Score ↑ |
|---|---|---|---|---|---|---|---|
| 10 | 65.82 | 71.99 | 37.21 | 0.550 | 7.21 | 0.029 | 58.54 |
| 100 | 65.83 | 72.19 | 37.16 | 0.550 | 7.22 | 0.029 | 58.64 |
| 1000 | 65.74 | 72.07 | 37.02 | 0.550 | 7.22 | 0.030 | 58.54 |
| 10000 | **67.88** | **72.12** | **37.37** | **0.557** | **7.28** | **0.035** | **59.16** |

Table 12: Ablation study on $\frac{\lambda_{\text{main}}}{\lambda_{\text{ref}}}$ for PROFIT on the DriveLM Benchmark. Higher values of $\frac{\lambda_{\text{main}}}{\lambda_{\text{ref}}}$ improves perception and prediction accuracy and language scores, showcasing the strength of PROFIT in generalizing to both VQA and scene understanding tasks.

### C.4.2 Implementation Details

We closely follow the training recipe from DriveLM contributors [2023]. For AdamW, we use a learning rate of $1e-3$ to fine-tune. PROFIT uses a learning rate of $1e-2$ for fine-tuning, while we set $\lambda_{ref}$ to $1/10000$ of the learning rate and $n_{ref} = 1$. We use 8 H100 GPUs for experiments.

### C.5 Visual Task Adaptation Benchmark

The Visual Task Adaptation Benchmark (VTAB-1K) Zhai et al. [2019] is a popular representation learning benchmark to evaluate generalization across a diverse set of image classification tasks. The 19 datasets can be grouped into 3 categories: Natural, Specialized and Structured groups. Each dataset contains 800 training examples and 200 validation examples. The domains for each dataset vary significantly, making it a challenging benchmark for PROFIT.

### C.5.1 Implementation Details

We follow the fine-tuning hyper-parameters, backbones, and classification heads parameters using the same setting as VPT Jia et al. [2022]. For experiments using PROFIT, we fine-tune the model with an initial learning rate of 0.01 and employ a cosine decay learning rate schedule, consistent with the approach in Jia et al. [2022]. We set $\lambda_{\text{ref}}$ to one-tenth of the learning rate and use $n_{\text{ref}} = 1$.

As shown in Table 13, we conducted PROFIT experiments in two different directions. One is to apply the PROFIT directly to fine-tune the models on the VTAB-1K dataset, this is denoted as "PROFIT w/o warmup" which yields poor performance, because the model starting point is poorly optimized for the target datasets. We also conducted another experiment, The "warmup" is conducted using the same AdamW optimizer and the same parameters as the FULL VPT described in the table 6 in Jia *et al.* Jia et al. [2022].

We also include in Table 13 comparisons against LoRA and NOAH, which are both popular current fine-tuning methods that allow for efficient fine-tuning. These methods both add additional trainable

| | Natural | | | | | | | Specialized | | | | Structured | | | | | | | |
|---|---|---|---|---|---|---|---|---|---|---|---|---|---|---|---|---|---|---|---|
| | Cifar100 | Caltech101 | DTD | Flower102 | Pets | SVHN | Sun397 | Camelyon | EuroSAT | Resisc45 | Retinopathy | Clevr-Count | Clevr-Dist | DMLab | KITTI-Dist | dSpr-Loc | dSpr-Ori | sNORB-Azim | sNORB-Ele |
| Full Jia et al. [2022] | 68.9 | 87.7 | 64.3 | 87.2 | 86.9 | 87.4 | 38.8 | 79.7 | 95.7 | 84.2 | 73.9 | 56.3 | 58.6 | 41.7 | 65.5 | 57.5 | 46.7 | 25.7 | 29.1 |
| Linear Jia et al. [2022] | 64.4 | 85.0 | 63.2 | 97.0 | 86.3 | 36.6 | 51.0 | 78.5 | 87.5 | 68.5 | 74.0 | 34.3 | 30.6 | 33.2 | 55.4 | 12.5 | 20.0 | 9.6 | 19.2 |
| VPT Jia et al. [2022] | **78.8** | 90.8 | 65.8 | 98.0 | 88.3 | 78.1 | 49.6 | 81.8 | **96.1** | 83.4 | 68.4 | 68.5 | 60.0 | 46.5 | 72.8 | 73.6 | 47.9 | **32.9** | 37.8 |
| LoRA Hu et al. [2022] | 67.1 | 91.4 | 69.4 | 98.8 | **90.4** | 85.3 | **54.0** | **84.9** | 95.3 | 84.4 | 73.6 | **82.9** | **69.2** | 49.8 | 78.5 | 75.7 | 47.1 | 31.0 | 44.0 |
| NOAH Zhang et al. [2025] | 69.6 | **92.7** | **70.2** | **99.1** | **90.4** | 86.1 | 53.7 | 84.4 | 95.4 | 83.9 | **75.8** | 82.8 | 68.9 | 49.9 | **81.7** | 81.8 | 48.3 | 32.8 | **44.2** |
| PROFIT | 58.9 | 55.6 | 51.4 | 92.4 | 50.8 | 19.6 | 37.7 | 79.3 | 53.8 | 43.5 | 73.5 | 12.6 | 37.9 | 21.1 | 77.5 | 10.2 | 24.6 | 23.6 | 10.7 |
| PROFIT (warmup) | 69.4 | 90.5 | 69.6 | 98.9 | 89.2 | **89.4** | 52.0 | **84.9** | 95.7 | **85.6** | 74.3 | 80.5 | 64.7 | **51.9** | 80.6 | **82.3** | **49.3** | 32.7 | 35.1 |

Table 13: Image classification accuracy on VTAB-1K. **Bold** = best, Underline = second-best. Ties for best are all **bolded**; no second-best is shown in such cases. We compare PROFIT against standard fine-tuning strategies, including full fine-tuning and linear fine-tuning (denoted by Full and Linear). A naive PROFIT violates the proximity condition and performs poorly, but with AdamW warm-up it achieves consistently higher accuracy.

weights to the model architecture, and PROFIT is competitive with these baselines. For future work we would be interested to see how to apply PROFIT on top of LoRA and NOAH implementations, as we suspect these methods would be synergistic and would further improve downstream model performance.

# D   Limitations

As discussed in Sec 3.2, PROFIT requires inference on $n_{\text{ref}} + 1$ batches (2 for $n_{\text{ref}} = 1$) per iteration, hence being slower than standard optimizers for fine-tuning. In practice, we observe that PROFIT converges earlier than corresponding baselines (which are trained longer for fair comparison), which may mitigate this. We also note that typical fine-tuning settings are shorter and not as involved as training from scratch, which may further alleviate this concern. Another limitation is that our method would consume slightly more memory as a result of instantiating two optimizers ($\mathbf{O}^{(\text{ref})}$ and $\mathbf{O}$).

## D.1   Memory Profiling

In training generative models, memory consumption is a bottleneck, and as mentioned in 3.4 the vanilla version of PROFIT induces some training-time memory overhead. For completeness we include those memory profiling experiments here. As model parameter counts and activation footprints grow, so does the GPU / VRAM requirement, often leading to training failures, batch-size reductions, and suboptimal hardware utilization. Memory profiling plays a critical role in assessing whether a proposed method remains practical under realistic resource constraints. In particular, when proposing a new optimizer like PROFIT, we must ensure that any additional memory overhead is manageable and that iteration time remains comparable. To validate the generality of our method, we profiled a diverse set of architectures that included both language and generative models.

We conducted a series of memory profiling experiments to compare PROFIT with the standard AdamW optimizer in a range of widely used architectures. These models were selected to represent different domains of interest for future research in fine-tuning large models. Table 14 summarizes the peak memory consumption and the per-iteration time for each setup.

| Model | Peak Mem. (MB) — AdamW | Peak Mem. (MB) — PROFIT | Iter. Time (ms) — AdamW | Iter. Time (ms) — PROFIT |
|---|---|---|---|---|
| OPT-1.3B Zhang et al. [2022] | 5532.5 | 6324.0 | 106.6 | 115.7 |
| OPT-2.7B Zhang et al. [2022] | 8602.4 | 10076.0 | 164.2 | 174.3 |
| LLaMA3-8B Dubey et al. [2024] | 20341.3 | 25693.4 | 474.7 | 490.4 |
| GPT-OSS-20B Agarwal et al. [2025] | 14502.5 | 17992.0 | 117.7 | 139.9 |
| Stable Diffusion v1-5 Rombach et al. [2022] | 5603.5 | 7134.5 | 130.3 | 142.9 |

Table 14: Peak memory usage and iteration time for AdamW vs. PROFIT. PROFIT shows modest additional memory overhead with comparable iteration latency. Note that we did not fine-tune the models with PROFIT for the accuracy here, and the experiments are for memory profiling only.

For OPT (Zhang et al. [2022]), LLaMA (Dubey et al. [2024]), and Stable Diffusion v1-5 (Rombach et al. [2022]), we adopt QLoRA (Dettmers et al. [2023]) for fine-tuning, while LoRA (Hu et al.

[2022]) is used for GPT-OSS-20B. This profiling highlights PROFIT's generality and efficiency across both language and vision–language models. As shown in Table 14, we have conducted the memory profiling experiments. The experiment shows that PROFIT is on par with AdamW, with an additional memory increase. This is because we treat both main and reference optimizers completely separately in order to better explore the hyperparameter space. As we discussed in the Section 3.4, it is expected that the memory overhead of the model training is increased approximately 25% over the AdamW (Loshchilov and Hutter [2019]) baseline.

## Impact Statement

The methods presented in this work represent general advancements in the field of machine learning, and are intended to be applicable in a wide range of machine learning applications. We hope that this work will help in preventing catastrophic forgetting and overfitting to potentially harmful data, which may be a positive ethical/societal effect. Other than that consideration, although there may be broader impact considerations in any downstream applications using our work, we feel that this work taken in isolation does not engender any additional societal impacts worth noting here.

