# OpenReview forum: "PROFIT: A Specialized Optimizer for Deep Fine Tuning"
_NeurIPS.cc/2025/Conference — NeurIPS 2025 poster_

### Official Review · Reviewer_97b1 · 2025-06-18

**Clarity:** 3
**Significance:** 3
**Originality:** 3
**Rating:** 5
**Confidence:** 2

**Summary:**

Fine-tuning pretrained deep models often sacrifices accuracy on the original task.
PROFIT is an optimizer that leverages the fact that a finetuning run starts from a converged model. By anchoring updates to the reference state and orthogonalizing gradients, PROFIT regularizes weight updates and yields better performance with minimal engineering overhead.

**Questions:**

In Section 4.3, could you also report zero-shot accuracy on a standard LLM benchmark suite (e.g. MMLU or LAMA) before and after fine-tuning to quantify how much general knowledge is retained?

**Ethical Concerns:**

["NO or VERY MINOR ethics concerns only"]

**Final Justification:**

I keep my score unchanged for the following reasons:
1. Authors gave a conceptual argument that PROFIT can be seen as an online variant of LWF and commit to add an empirical comparison in the camera-ready. The current comparisons already cover the optimizers that make the same resource assumptions, and the theoretical clarification shows why LWF is disadvantaged.
1. Authors promised MMLU results in the final version.

**Limitations:**

yes

**Paper Formatting Concerns:**

No major formatting issues found

**Quality:**

3

**Strengths And Weaknesses:**

**Strengths**
1. PROFIT can be easily dropped into an existing deep learning framework and it avoids the overhead of adapters or LoRA.
1. Evaluations cover images, vision-language, and motion trajectories. Waymo results are convincing.
1. When the pre-train and fine-tune distributions diverge, authors present an AdamW warm-up fix.
1. Theoretical analysis explains why the projection step limits weight drift.

**Weaknesses**
1. Although the authors mention that "Since our method is an optimizer by design and does not require additional storage (unlike the continual learning paradigm), we do not benchmark against continual learning methods.", it would be great to see LWF(Li & Hoiem 2017) included as a baseline, since it is a frozen model copy with soft target distillation. This would give more context to PROFIT's gains.

**Minor**
1. line 187: occrs -> occurs
1. line 189: oprtimizer -> optimizer
1. line 315: during fine-tuning. by leveraging -> during fine-tuning by leveraging
1. labels in figures are hard to read, especially in Table 2 and Figure 2. Consider splitting them or move some parts to the appendix.

---

> ### Author Rebuttal · Authors · 2025-07-31
>
> Thank you for your positive assessment of our work. We are very much aligned that the positives you pointed out (our convincing results and our method’s lightweightness) are principles at the heart of what we hoped to accomplish with PROFIT.
>
> > Including LWF as a baseline
>
> We will plan to run an analysis against LWF for a revision, but for now can offer the following theoretical analysis: for each step of PROFIT, we update the model by two signals: the signal $-\Delta$ which is equivalent to the soft distillation target update within LWF, and an additional update $g\perp \Delta$ which is a regularized version of the standard new task update within LWF. Therefore, one can think of PROFIT as a flexible version of LWF where we constantly update the “frozen” model to the model state at $t-1$. This allows PROFIT to surpass a few notable limitations to LWF that limited its practical utility:
>
> 1. LWF requires a separate inference step on a model copy that must be kept around.
> 2. LWF is difficult to configure when a model has to be maintained through multiple fine-tuning iterations, as it is unclear what to use as the frozen “base” model through multiple steps. PROFIT always sets the frozen model to the model at timestep $t-1$, which removes ambiguity on frozen model choice.
>
> In addition, since PROFIT is a drop-in optimizer, it is significantly easier to configure and run.
>
> Ultimately, it is reasonable to think of PROFIT as a practical evolution of LWF which removes standard barriers and ambiguity, while preserving the core idea. Importantly, PROFIT accomplishes this while removing assumed access to material data (i.e. frozen model weights) that LWF relies on. As such, we originally felt the comparison would be not too instructive, similarly to how comparing LWF to fine-tuning a model with the original dataset intact would also be difficult to interpret. Comparison to standard optimizers which make the same set of assumptions on access to data is more apples-to-apples. However, we are aware that given how much our methodology is inspired by LWF, a comparison may be of interest and we are working hard on completing that comparison to include in the supplementary. We hope that in the meantime this additional context at least clarifies how PROFIT is theoretically positioned with respect to LWF and why we decided to omit the comparison in the main paper to keep the analysis clean.
>
> > Zero-shot accuracy on a standard LLM benchmark suite
>
> We will plan to run experiments on MMLU for the camera-ready version. One benchmark we ran that provides insight into the amount of general information retained are the CIFAR baselines we present in Section 4.1. Although not an LLM baseline, these experiments show that PROFIT beats all other methods in terms of knowledge retention and performance on the transfer set. We note that PROFIT is not intended as a silver bullet against catastrophic forgetting, but is mainly designed to mitigate forgetting in such a way that optimizes learning on the transfer set. This design principle explains why we largely reported transfer set accuracy results. However, we agree that these results could provide additional context on another baseline for PROFIT and will begin to run those experiments.

---

> > ### Comment · Reviewer_97b1 · 2025-08-04
> >
> > I thank the authors for the detailed responses to my questions. My score remains unchanged.

---

### Official Review · Reviewer_Fqcd · 2025-07-03

**Clarity:** 2
**Significance:** 2
**Originality:** 3
**Rating:** 4
**Confidence:** 4

**Summary:**

This paper introduces PROFIT (PROximal Fine Tuning), a novel optimizer designed to mitigate catastrophic forgetting during the fine-tuning of pre-trained models. The core idea is to reframe the optimization process as a temporal multi-task learning problem, where the model must learn a new task while not forgetting its original one. PROFIT achieves this by treating the model's pre-fine-tuning state as an "anchor." It then uses gradient orthogonalization to adjust the update steps for the new task, specifically nullifying any component of the gradient that directly conflicts with returning to the anchored state. The paper positions PROFIT as a better alternative to standard full-model fine-tuning using optimizers like AdamW. Experiments across various domains, including image classification and motion prediction, demonstrate that PROFIT improves performance over standard baselines.

**Questions:**

In Table 2, the paper includes results from several state-of-the-art parameter-efficient methods (e.g., VPT, NOAH) which often outperform  PROFIT (warmup). Since the paper's text explicitly defines its comparison against standard Full and Linear fine-tuning , could you clarify the intended role of these other SOTA methods in your analysis? What is the key takeaway for a reader when comparing PROFIT to these methods?

PROFIT (warmup) strategy is presented as an effective solution for the non-proximal VTAB benchmark . Was the choice of a 10-epoch AdamW warmup a result of a systematic hyperparameter search, or was it a heuristic? How sensitive is the final performance to the duration and the specific optimizer used for this warmup phase?

I will be happy to adjust score if the author can address my concern (plus appropriate response to the weakness).

**Ethical Concerns:**

["NO or VERY MINOR ethics concerns only"]

**Final Justification:**

Based on the authors' response, I feel the problem of having another copy of optimizer would be a bit of waste, considering the marginal improvement. However, the authors have provided evidence this can be avoided after making the decision of using certain types of optimizers. I am happy to raise score to borderline accept.

**Limitations:**

Yes

**Quality:**

3

**Strengths And Weaknesses:**

Strength
+ Paper is well writen.
+The core idea of framing fine-tuning as a temporal multi-task problem and using gradient orthogonalization as a mechanism is reasonable and presents an interesting approach to the challenge of catastrophic forgetting.

+The paper's central claim—that PROFIT outperforms standard full-model fine-tuning—is supported by empirical results across a decent breadth of tasks, including image classification, visual task adaptation, and motion forecasting.

Weakness
- The presentation in Table 2, which compares PROFIT to many advanced optimizers, could be clearer. The paper does not fully disclose the reason for including results from several state-of-the-art methods (e.g., VPT, NOAH) that often show superior performance. While the text specifies that its direct comparison is to standard fine-tuning, the table's structure may confuse readers about the paper's positioning relative to the broader SOTA landscape.

- The paper claims PROFIT is easy to integrate but lacks a comparative analysis of implementation complexity. It is unclear how the effort to implement PROFIT compares to setting up the other advanced optimizers listed, particularly parameter-efficient techniques like LoRA or VPT which also aim to simplify model adaptation

- The paper should disclose the spatial (memory) overhead. Storing reference weights for large-scale models could have significant memory implications, which is a crucial factor for practical deployment, but this is not measured or discussed.

---

> ### Author Rebuttal · Authors · 2025-07-31
>
> Thank you for finding our paper “well written” and “supported by empirical results across a decent breadth of tasks”.
>
> > Was the choice of a 10-epoch AdamW warmup a result of a systematic hyperparameter search?
>
> The VTAB strategy was a heuristic chosen to warmup for 10% of epochs and fine-tune for the remaining epochs. We tried some variants with slightly higher warm-up epochs and found similar performance. Therefore, when a practitioner is faced with finetuning on arbitrary distributions, we recommend starting with a warmup for 10-20% of the total epochs.
>
> > Paper does not fully disclose the reason for including results from several state-of-the-art methods (e.g., VPT, NOAH)
>
> This is an astute observation. We include these results for completeness and to show that PROFIT is competitive with these SOTA methods. However, we emphasize that these methods induce additional trainable parameters and the comparison is not apples-to-apples. We will add a column on Table 2 to indicate when a method induces additional trainable parameters, format the table to show clearer grouping between different methodology classes, and also include discussion of this point in the main text.
>
> > How the effort to implement PROFIT compares to setting up other advanced optimizers
>
> We outline the algorithm of PROFIT in Alg 1. This logic can be encapsulated in an PyTorch optimizer, and our implementation is ~20 lines of code.
>
> > Paper should disclose the spatial (memory) overhead
>
> We discuss the memory complexity for our method in Table 8 of the supplementary. Storing an additional copy of states for the reference optimizer does increase memory complexity, which is a limitation of our method as we point out in Sec 2.2.6 of the supplementary.
>
> While PROFIT is implemented with two optimizers in our experiments for generality and flexibility, this is not a requirement. In practice, one can reuse a single optimizer instance (e.g., SGD, AdamW, etc.) for both the main and reference model, provided care is taken to isolate their parameter updates. This reduces the additional memory footprint to effectively zero.
>
> Moreover, PROFIT is compatible with parameter-efficient fine-tuning techniques such as adapters (e.g., LoRA, VPT). In Table 3 of the supplementary, we show that combining PROFIT with these methods yields consistent performance improvements while keeping memory usage low.

---

> ### Comment · Reviewer_Fqcd · 2025-08-04
> **Follow up on the gpu memory / computing time overhead**
>
> Thanks the author for the response. Table 3 in supplementary materials is insightful. I would like the authors to further explain to the unclear points in their response: The authors first mentioned "Storing an additional copy of states for the reference optimizer does increase memory complexity, which is a limitation of our method as we point out in Sec 2.2.6 of the supplementary." Is this inevitable? Then, the authors mentioned "This reduces the additional memory footprint to effectively zero." Is this saying the current implementation is further costing more gpu memory? Please verify this.
>
> Nevertheless, I would like to suggest the authors provide a typical memory / time usage profiling for the typical tasks including diffusion modeling, larger language model such as models from 1B to 7B, under the setting of parameter-efficient fine-tuning. This will better help to understand the wider applicability of the proposed method. Note that, this is not a suggestion asking to provide results showing improved performance, but asking the profiling results to reveal the potential gpu memory cost / computing time analysis.

---

> ### Author Response · Authors · 2025-08-09
>
> We thank the reviewer for considering our response. In terms of memory benchmarking, we noted earlier that our current implementation of PROFIT induces additional memory footprint because we treat both main and reference optimizers completely separately in order to better explore the hyperparameter space. In theory, it is possible to zero out the memory footprint by implementing PROFIT logic in the same optimizer instance. We went back and implemented this version, where the optimizers do not make separate copies of the model weight parameters but share a copy, and we show the memory footprint results below:
>
> | Models                                            | Peak Memory in MB (AdamW) | Peak Memory in MB (PROFIT – vanilla implementation) | Peak Memory in MB (PROFIT — consolidated optimizer) |
> |---------------------------------------------------|--------------------------|---------------------------|--------------------------------------------------------|
> | OPT-1.3B                                 | 5532.5                   | 6324.0                    | 5531.1                                                 |
> | OPT-2.7B                                 | 8602.4                   | 10076.0                   | 8601.4                                                 |
> | Llama3-8B                        | 20341.3                  | 25693.4                   | 20341.3                                                |
> | gpt-oss-20b                                | 14502.5                  | 17992.0                   | 14502.5                                                |
> | Stable Diffusion v1-5                   | 5603.5                   | 7134.5                    | 5602.5                                                 |
>
>
> As we see, the memory footprint is now effectively zero. At baseline, PROFIT induces zero additional cost at inference, and we demonstrated now that it also can have zero footprint at training. The main reason we did not do this during our experiments is that a separate custom implementation has to be done for every (O_ref, O_main) pair, and so it was an impediment for hyperparameter search. However, once you know the right optimizer types (and O_ref should generally be SGD), this implementation can be used to fully eliminate memory overhead. We hope this clarifies this particular point about why we felt it was justified to tolerate the additional memory used in our experiments, and alleviates the reviewer's concern.

---

### Official Review · Reviewer_4J8q · 2025-07-03

**Clarity:** 3
**Significance:** 2
**Originality:** 2
**Rating:** 3
**Confidence:** 4

**Summary:**

This paper proposes PROFIT, a specialized optimizer designed to improve fine-tuning of pre-trained deep learning models. Unlike conventional optimizers such as SGD or Adam, which are designed for training from scratch, PROFIT leverages the structure of an already converged model by treating its current state as a reference point. By framing fine-tuning as a form of temporal multi-task learning, PROFIT enables accurate and efficient adaptation to new but related tasks without introducing additional parameters.

**Questions:**

- Given this proximal setting, why is a specialized optimizer like PROFIT necessary instead of simply applying standard proximal optimization techniques (e.g., L2 regularization around the pre-trained weights)? A direct comparison or ablation would help clarify whether the performance gain stems from the temporal orthogonalization approach itself or merely from staying close to the initial weights.
- PROFIT introduces new hyperparameters (e.g., reference steps $n_\text{ref}$, $\lambda_\text{ref}$), which influence the computation cost and optimization behavior. What is the actual impact of $n_\text{ref}$ on performance and forgetting? An ablation study varying $n_\text{ref}$ (e.g., 1, 2, 4, 8…) would help determine whether the reference optimization phase is truly beneficial or just an overhead.
- The proposed method assumes a proximal relationship between pre-training and fine-tuning data distributions, which is central to its effectiveness. However, in many real-world applications, this assumption may not hold strictly. Could the authors provide more concrete and diverse examples where proximal fine-tuning naturally arises in practice? Some clarification would help assess the general applicability of the method beyond narrowly defined scenarios like autonomous driving or sensor-specific tasks.
- The current method focuses on proximal fine-tuning, where pre-training and fine-tuning distributions are assumed to be similar. Can the authors clarify whether PROFIT could be extended or adapted for domain adaptation scenarios, where the source and target distributions differ moderately or significantly?
- Figure 1 appears to be central in conveying the intuition behind PROFIT’s update mechanism. However, the diagram’s intent and the plots are not clearly explained and may be confusing to readers. Could the authors revise or elaborate on the figure to more explicitly show how PROFIT differs from standard optimizers in both conceptual and operational terms?

**Ethical Concerns:**

["NO or VERY MINOR ethics concerns only"]

**Final Justification:**

I appreciate the authors’ detailed and thoughtful response to my questions. While several points were clarified, some of my core concerns remain unresolved. Specifically, the method is framed from a proximal optimization perspective, yet no comparison is provided against a standard L2-based proximal optimization baseline, which could be a natural and principled reference in this context. Including such a baseline is crucial to properly assess the contribution of the proposed approach.

**Limitations:**

Yes

**Paper Formatting Concerns:**

1. The submitted paper content:  No issue
2. Paper references: Format should be fixed
3. The NeurIPS paper checklist: No issue

**Quality:**

2

**Strengths And Weaknesses:**

1. Strengths
- PROFIT is one of the first optimizers explicitly designed for fine-tuning pre-trained (converged) models, addressing a key gap where traditional optimizers like SGD and Adam fall short.
- The use of orthogonal projection to decouple new-task gradients from the restoration direction (toward the original state) is a simple and principled strategy inspired by multi-task learning, allowing stable and targeted adaptation.

2. Weaknesses
- PROFIT is only effective when the fine-tuning dataset is from a distribution similar to the pre-training dataset. It fails or performs poorly in cases of significant domain shifts, limiting its general applicability.
- PROFIT introduces a reference optimization phase ($n_\text{ref}$ steps) for each training iteration, which can increase training time, although this may be offset by faster convergence.
- Please also refer to the following questions.

---

> ### Author Rebuttal · Authors · 2025-07-31
>
> Thank you for your positive reading and recognizing our work as “one of the first optimizers explicitly designed for fine-tuning pre-trained deep learning models”.
>
> > How PROFIT compares to standard proximal optimization techniques (e.g: L2 regularization around the pre-trained weight)?
>
> An L2 regularizer around the pre-trained weights relies on a maximally uninformative Bayesian prior that will indiscriminately pull updates back towards the pre-trained point. However, L2 regularization does not take into account weight updates towards the fine-tuning task at all. We already know that such indiscriminate preference of the original convergence point is suboptimal, because in all fine-tuning/post-training settings the old convergence point is assumed to be a poor solution (otherwise we would not fine-tune at all). PROFIT resolves this issue by continuing to update the model “anchor” point after each step, which allows for dynamically useful updates throughout model training.
>
> We also note that L2 regularization around the pre-trained weight would necessitate keeping a copy of the model weights around to be able to perform this regularization step. This both induces compute/memory overhead but also adds ambiguity when a model needs to go through multiple iterative stages of fine-tuning. L2 regularization also adds a magnitude hyperparameter while PROFIT handles update magnitude balancing automatically and with theoretical grounding. These restrictions make this type of L2 regularization scheme less practical. PROFIT does not require any additional data, which is why the comparison itself is less meaningful (just like it would be difficult to compare the L2 regularization algorithm to a situation where we have access to all the original dataset). We chose to primarily focus on comparisons between PROFIT and settings that assume access to the same cardinality of information to ensure comparisons are all apples-to-apples.
>
> Thus, we do not believe L2 regularization is an effective proximal optimization technique but also is only possible in a privileged setting that assumes higher levels of information access than we have in the more general PROFIT setting. To the best of our knowledge, we are the first to examine the “proximal fine-tuning” setting with these data restrictions, which also is the most general setting for fine-tuning.
>
> > What is the actual impact of hyper-parameters on performance and forgetting?
>
> We quantify the impact of hyper-parameters in the supplementary. In particular, we point the reviewer to Table 4 and 5 where we ablate the critical hyper-parameters of our method. We leverage these hyperparameters in all the discussed experimental settings.
>
> > Examples where proximal fine-tuning arise in practice
>
> The proximal setting is very common in machine learning applications. In computer vision, this can be applied when the sensor modality stays constant through the lifecycle - expanding classification/segmentation taxonomies in self-driving, localization and action in new indoor environments for humanoid robots, new rooms/layouts/maps for AR/VR applications. In language models, this can be used to fine-tune a base model to support new tool-calls or function calls, or in post-training to ensure pre-trained knowledge isn’t forgotten while instruction-following capabilities are being introduced. In general, any setting where a model has to be updated due to collection of new data would also fall into the proximal setting.
> We note that even nonproximal fine-tuning settings can be made proximal with a mild warmup stage. In Section 4.2, we presented experiments on VTAB where naive application of PROFIT leads to poor results, but a simple warmup period (10 out of 100 epochs) transforms the VTAB fine-tuning setting into a proximal setting, and at that point PROFIT leads to significantly improved results. In other words, even non-proximal fine-tuning settings can be easily transformed into proximal settings, making PROFIT actually a fully general algorithm for fine-tuning.
>
> > Whether PROFIT could be extended where source and target distributions differ
>
> We point the reviewer to Section 4.2, where we show our method can be mitigated in practice by using a short warm-up phase with standard fine-tuning before switching to PROFIT. Furthermore, we show this approach works well in Table 2.
>
> > Revise or elaborate on Figure 1
>
> We discuss how PROFIT differs from standard optimizers in the caption of Figure 1. Standard optimizers which take successive steps through parameter space, while the reference optimizer (superscript “r” - for “reference” - in the figure) keeps track how the system tries to move away from good state ($x_0$). The good state is then restored ($x_0$) and then reference steps are translated to $x_0$. The main optimizer then takes a step and the gradient update is orthogonalized wrt to the translated reference vector. For further details, we point the reviewer to Alg 1 and Sec 3.1 where we elaborate the intuition and operational mechanism behind our method.

---

> ### Comment · Reviewer_4J8q · 2025-08-05
>
> Thank you for your detailed and thorough rebuttal. I appreciate the time and effort you put into addressing all of my questions and concerns. However, I still have a few additional questions and points of clarification that I would like to raise.
>
> - The claim that L2 regularization around the pre-trained weights is “maximally uninformative” is debatable, as it in fact corresponds to an informative Gaussian prior when centered on pre-trained parameters. While it indeed pulls updates indiscriminately toward the anchor point, this can be beneficial for stability and preventing overfitting. The assumption that the pre-trained point is always a poor solution is also questionable, as fine-tuning often serves to adapt an already strong model rather than replace it entirely.
> - In addition, since PROFIT appears to operate on the basis of full fine-tuning, it would be important to clarify whether similar benefits could be achieved in combination with parameter-efficient fine-tuning (PEFT) methods.

---

> > ### Author Response · Authors · 2025-08-05
> >
> > Thank you for these questions, and we are happy to clarify these further:
> >
> > > Pre-trained point may provide a good solution
> >
> > Using the pre-trained point as a good solution for the fine-tuned task may not always apply. For instance, using zero-shot LLAMA-Adapter-v2 on DriveLM benchmark shows:
> >
> > | Model                | Accuracy (%) ↑ | GPT Score (%) ↑ | Match Score (%) ↑ | BLEU-4 ↑ | ROUGE_L ↑ | CIDEr ↑ | Final Score ↑ |
> > | -------------------- | -------------- | --------------- | ----------------- | -------- | --------- | ------- | ------------- |
> > | Zero-shot            | 0              | 67.75           | 18.83             | 0.011    | 0.20      | 0.007   | 32.84         |
> > | PROFIT fine-tuning   | 62.21          | 70.57           | 36.33             | 0.541    | 7.16      | 0.023   | 56.98         |
> >
> > The zero-shot model is essentially evaluated at the anchor point. This model is significantly worse at perception and prediction accuracy, while also struggling in all VQA metrics. We point the reviewer to Sec 2.4 of the supplementary for a detailed explanation on these metrics.
> >
> > This result provides further evidence regarding the reviewer’s question on whether performance gain come from PROFIT as opposed to simply staying at the initial weights. Unless we provide downstream supervision (i.e, fine-tune) for this task, we are unlikely to get far on DriveLM. While we are happy to provide a comparison with L2 regularizer in our final manuscript, we firmly believe this baseline will perform similarly (if not worse) to the zero-shot baseline.
> >
> > > Compatibility with PEFT methods
> >
> > We have conducted extensive additional experiments with adapters including LoRA, Visual Prompt Tuning, and ConvAdapter, and provide these results in Table 3 of the supplementary. Choosing PROFIT as the optimizer consistently provides positive benefit over using an Adam optimizer, showing that PROFIT is complementary to adapter methods and can be used in parallel. These results indicate that PROFIT can serve as a general optimization layer that can be added to diverse fine-tuning settings.

---

> ### Comment · Reviewer_4J8q · 2025-08-07
>
> Thank you for your detailed response to my questions. I appreciate the clarifications you provided. However, some of my concerns remain unaddressed. Since your proposed method is motivated from a proximal optimization perspective, I believe that a comparison against a standard L2-based proximal optimization baseline is particularly important. Accordingly, I will make a slight adjustment to my score while keeping it generally consistent.

---

> ### Author Response · Authors · 2025-08-08
>
> We thank the reviewer for going through our response. We believe the above table, experiments in Sec 4.1, 4.2, 4.3, 4.4, and the supplementary showcase the effectiveness of our method as compared to fine-tuning with another method. We are happy to add this baseline to our camera-ready version if given the chance to proceed.
>
> We are happy to engage in further discussion to clarify additional concerns.

---

### Official Review · Reviewer_FYXs · 2025-07-07

**Clarity:** 3
**Significance:** 3
**Originality:** 3
**Rating:** 4
**Confidence:** 3

**Summary:**

This abstract presents PROFIT, a novel optimizer targeting improved model accuracy for fine-tuning pre-trained models, addressing a perceived shift in the field towards efficiency over accuracy. The authors position PROFIT as one of the first optimizers specifically designed for incrementally fine-tuning models that have already converged on prior tasks/datasets.

**Questions:**

See above

**Ethical Concerns:**

["NO or VERY MINOR ethics concerns only"]

**Final Justification:**

Thanks to the authors. I will keep my score.

**Limitations:**

See above

**Paper Formatting Concerns:**

See above

**Quality:**

3

**Strengths And Weaknesses:**

Strengths:

1. PROFIT uniquely resolves catastrophic forgetting by projecting new-task gradients onto a plane orthogonal to pretrained model displacements

2. The method integrates as a plug-and-play optimizer wrapper requiring no architectural changes or extra parameters.

3. Demonstrates consistent gains in image classification, autonomous driving VQA, and motion prediction .

Weaknesses:

1. The caption of Table 2 states that bold values indicate the best results, yet there are ​​two bold entries per column​​. Most of the proposed method’s results are bolded, which is confusing.

2. Table 2 compares the proposed method with classic PEFT (Parameter-Efficient Fine-Tuning) algorithms, but ​​image classification alone does not sufficiently demonstrate PEFT’s applicability to general vision tasks​​. For real-world applications (e.g., autonomous driving, robotics), ​​more complex tasks like object detection or instance segmentation​​ are more practical and have been benchmarked in recent works [1-2]. Given this is NeurIPS, I suggest the authors to ​​expand experiments to validate the method’s generality​​.

[1] 5% > 100%: Breaking Performance Shackles of Full Fine-Tuning on Visual Recognition Tasks

[2] 1% vs 100%: Parameter-Efficient Low-Rank Adapter for Dense Predictions

3. Lines 172, 183, 185, 194, etc., exhibit ​​encoding artifacts​​ at their line endings.

---

> ### Author Rebuttal · Authors · 2025-07-31
>
> Thank you for your thoughtful review and recognition that our work “uniquely resolves catastrophic forgetting” and “integrates as a plug-and-play optimizer”.
>
> > Two bolded entries per column in Table 2
>
> We include results from adapter methods such as VPT, NOAH, etc for completeness and to show that PROFIT is competitive with these SOTA methods. However, we emphasize that these methods induce additional trainable parameters and the comparison is not apples-to-apples. We will add a column on Table 2 to indicate when a method induces additional trainable parameters, format the table to show clearer grouping between different methodology classes, and also include discussion of this point in the main text.
>
> > Suggest the authors to ​​expand experiments
>
> We thank the reviewer for the suggestion. We believe the experiments reported in Sec 4.1, 4.2,  4.3, and 4.4  are comprehensive, and although we are time-constrained and cannot run the requested experiments in time for this response, we are confident they will show the same trend as the other experiments presented. We note that benchmarks added for VTAB are a wide variety of classification tasks (Sec 4.2), and we also demonstrated positive results on VLMs in Sec 4.3, which represents a bleeding-edge modern computer vision task. Our experiments on DriveLM also show improvements in a setting that is tangential to computer vision and very high-dimensional. However, we will plan to add additional experiments in time for the camera-ready.

---

### Official Review · Reviewer_HVkg · 2025-07-09

**Clarity:** 2
**Significance:** 2
**Originality:** 2
**Rating:** 3
**Confidence:** 3

**Summary:**

The paper proposes a novel optimization algorithm for “proximal fine-tuning” where fine-tuning dataset is from a similar distribution to pre-training. The objective of this work is to fine-tune a fully pre-trained model to perform well on both fine-tuning and pre-training tasks. To this end, it employs two optimizers: a standard optimizer and reference optimizer. The proposed algorithm first uses reference optimizer to “explore” loss landscape and updates the standard optimizer using orthogonal projection of the gradient onto the change of parameters in the reference optimizer, not to catastrophically forget the parameters learned from pre-training dataset. The paper provides extensive experimental results on  image classification, visual question answering and robotics motion prediction tasks on datasets that does and does not satisfy the proximal fine-tuning assumption.

**Questions:**

* What is the difference between Section 4.1 and 4.2? I do not see if this distinction is necessary.
* Can the authors provide the formula for the orthogonal projection explicitly for better readability?

**Ethical Concerns:**

["NO or VERY MINOR ethics concerns only"]

**Final Justification:**

The authors’ rebuttal addresses some of my initial concerns. However, I remain unconvinced that the proximal assumption: the fine-tuning dataset comes from a distribution similar to that of pre-training, is generally realistic or justified in practical scenarios. In addition, the results are based on a single random seed, which weakens the credibility of the findings, particularly when performance differences between methods are relatively small.
Finally, while the writing of the paper needs improvement, I did not observe substantial effort or willingness to address this in the revision. Taken together, I do not believe this work meets the acceptance standards of NeurIPS.

**Limitations:**

Limitations of the methods related to assumptions and additional hyperparamters were noted and discussed in Section 3.3 and 3.4.

**Paper Formatting Concerns:**

I do not have formatting concerns.

**Quality:**

2

**Strengths And Weaknesses:**

**Strengths**
1. The paper proposes a simple but effective method tailored for fine-tuning.
2. The assumptions (which can be limitations) were clearly noted and discussed in the paper to reduce any potential confusion readers may have.
3. It provides some theoretical derivation that supports the effectiveness of the proposed method.
4. It provides extensive experimental results on various fine-tuning tasks.


**Weaknesses**
1. Some parts of the writing lack clarity.
	1.  In line 154-156, it is not clear how “because…” is a justification for projecting g onto $\Delta$ not the other way around.
	2. In line 157-158, “take both steps…” and “to take a step” sound like $\theta$ is updated three times in total but it seems like it is updated once after projection.
	3. Minor: the term equilibrium was used without sufficient explanation, making its meaning unclear for a while.
2. The proximal fine-tuning assumption could be a strong assumption. Although the authors explicitly mentioned this and provided an example of perception in autonomous driving, there are obviously more cases where this assumption does not meet than the cases where it does. To have this assumption satisfied for more cases, the pre-training dataset needs to be large enough to be similar to any potential fine-tuning datasets.
3. In addition, it is not clear how to measure if the proximal fine-tuning assumption is satisfied or not between pre-training and fine-tuning datasets.
4. It is not clear why the standard SGD is recommended for $O^\text{ref}$. Line 208-209 sounds like it is solely for the sake of mathematical simplicity. Is it a correct interpretation? Can the authors provide more fundamental reason?
5. Experiments are done with only one random seed. Without standard deviation, it is hard to know how significant differences are.
6. Bold in Table 2 does not seem to be for “the highest accuracy are highlighted in bold.” which may mislead readers.
7. The 10 epoch warmup with AdamW seems like a heuristics based on extensive experiments. How should practitioners choose the number of epochs for an arbitrary pair of pre-training and fine-tuning datasets?
8. The authors provided two justifications for additional computation using $n_\text{ref}$ in Section 3.3. However, I cannot find experiment results that support these claims.
9. There is no formal proof for Theorem 3.1 (I couldn’t find any proofs in the appendix). Could the authors provide the formal proof for Theorem 3.1? In particular, I do not really understand “If the learning rate is sufficiently small, then $\nabla_{\theta’} L_\text{old}$  must point in the $\Delta$ (as defined in Alg 1) direction”

---

> ### Author Rebuttal · Authors · 2025-07-31
>
> Thank you for your review and finding our method “simple and effective” with “extensive experimental results”.
>
> > If proximal fine-tuning assumption could be a strong assumption
>
> The proximal setting is very common in machine learning applications. In computer vision, this can be applied when the sensor modality stays constant through the lifecycle - expanding classification/segmentation taxonomies in self-driving, localization and action in new indoor environments for humanoid robots, new rooms/layouts/maps for AR/VR applications. In language models, this can be used to fine-tune a base model to support new tool-calls or function calls, or in post-training to ensure pre-trained knowledge isn’t forgotten while instruction-following capabilities are being introduced. In general, any setting where a model has to be updated due to collection of new data would also fall into the proximal setting.
> We note that even nonproximal fine-tuning settings can be made proximal with a mild warmup stage. In Section 4.2, we presented experiments on VTAB where naive application of PROFIT leads to poor results, but a simple warmup period (10 out of 100 epochs) transforms the VTAB fine-tuning setting into a proximal setting, and at that point PROFIT leads to significantly improved results. In other words, even non-proximal fine-tuning settings can be easily transformed into proximal settings, making PROFIT actually a fully general algorithm for fine-tuning.
>
> > How to measure if the proximal fine-tuning assumption is satisfied
>
> In our setup, the pre-training and fine-tuning datasets are known to be sampled from the same underlying distribution. We also do show situations where proximality fails - for example in VTAB (Sec 4.2). However, we also noted within our VTAB experiments that a short warmup cycle (10 out of 100 epochs) moves the VTAB setting into the proximality range and at that point PROFIT provides significant training benefits. While we find measurement of the proximality condition to be complex and very dependent on dataset and model properties, we at least provide a method to induce proximality in any setting which should work for general use-cases.
>
> > How to choose the number of epochs for an arbitrary pair of pre-training and fine-tuning datasets
>
> The VTAB strategy was a heuristic chosen to warmup for 10% of epochs and fine-tune for the remaining epochs. We tried some variants with slightly higher warm-up epochs and found similar performance. Therefore, when a practitioner is faced with finetuning on arbitrary distributions, we recommend starting with a warmup for 10-20% of the total epochs.
>
> > Why the standard SGD is recommended
>
> The reference optimizer is used to perturb the model away from the local optima to estimate the local loss landscape. By construction, $\Delta$, the pseudo-gradient (L155), minimizes the loss on the old task. Using optimizers which rely on other statistics would mean that $\Delta$ does not faithfully represent the direction to minimize old task loss, since it may be influenced by information gathered at another point in the loss surface. This property only holds true for vanilla SGD, and hence the reference optimizer should be set to SGD.
>
> In other words, PROFIT’s core algorithm is powered by simple orthogonality relationships between the main and reference gradient updates. If the reference update is calculated by any other optimizer than SGD, it is too complex for the simple orthogonality calculus to still hold true.
>
> > Experiments are done with only one random seed
>
> Experiments were run multiple times and we report error bars here. These will be added in a revision if we are allowed to proceed. CIFAR experiments have a standard error <= 0.01, DriveLM experiments have standard error within 0.2 on accuracy, VTAB experiments shows the mean accuracy over 3 repeats (as discussed in the VTAB paper), Waymo Open Motion Dataset experiments have standard errors within 0.005m for 3s metrics and 0.015m for 8s metrics.
>
> > Formal proof for Theorem 3.1
>
> At a convergence point $\mathbf{x}_0$, the first-order gradient vanishes and to leading order the loss surface will look like $L(\mathbf{x}) \approx \mathbf{x}_0 + 0.5(\mathbf{x} - \mathbf{x}_0)^t\mathbf{H}_0(\mathbf{x} - \mathbf{x}_0)$ for positive definite hessian $\mathbf{H}_0$. The gradient within this region is therefore $\nabla L(\mathbf{x}) \approx \mathbf{H}_0(\mathbf{x} - \mathbf{x}_0) \equiv \mathbf{H}_0\Delta$. Because $\mathbf{H}_0$ is positive definite given that $\mathbf{x}_0$ was a critical point, this means that $(-\Delta)^t\mathbf{H}_0\Delta < 0$, which means the $-\Delta$ direction is a valid gradient descent direction. We apologize for the confusion - by “$a$ points in the direction of $b$” in the part of the proof you referenced we intended the meaning to imply “pointing in the same direction as,” which implies $\langle a, b\rangle > 0$ and shows that $\Delta$ is a valid gradient descent direction. We hope this more formal proof clarifies any ambiguity and will include it in the final version of the paper.
>
> > Difference between Sec 4.1 and Sec 4.2
>
> In Sec 4.1, we demonstrate results on image classification when the pre-training and fine-tuning datasets are sampled from the same distributions. In Sec 4.2, we illustrate a case where this assumption may not hold, and demonstrate a strategy to use PROFIT in such cases.

---

### Note · Authors · 2025-08-13

We once again would like to thank all reviewers and the AC for considering our work for publication at NeurIPS 2025. We enjoyed the spirited discussion, and feel encouraged by the general assessments that our PROFIT algorithm has significant general utility and tackles a problem (iterative fine-tuning) that has become core to the machine learning community. We also appreciate the myriad requests for additional experiments and the interest in seeing PROFIT’s performance in diverse settings. We plan to address these requests in time for the camera-ready, but also feel enthused at the sheer number of diverse settings the reviewers brought up as additional testing grounds for PROFIT - we hope that at a high-level this demonstrates how potentially broadly applicable our algorithm truly can be.

In general, we believe we have demonstrated in our manuscript that a single algorithm is able to provide noteworthy performance gains in a broad number of settings, from large-scale computer vision to deep sequence prediction to modern LLMs. Importantly, PROFIT also makes significantly fewer assumptions on training settings than most comparable fine-tuning algorithms; it needs zero history of data or model weights and can be implemented in an optimizer that can be immediately dropped into any system. We feel strong conviction that the sheer ease of use and number of potential applications of PROFIT will make it of great interest to the machine learning community, and that it can potentially become a standard optimization tool in the ML practitioner’s arsenal.

We once again thank all reviewers, ACs, and conference organizers for their consideration.

---

### Decision · Program_Chairs · 2025-09-17

**Decision:**

Accept (poster)

**Comment:**

This is a nice paper that introduces a novel and practical optimizer, PROFIT, specifically designed for the ubiquitous task of fine-tuning pre-trained models. I am pleased to recommend this paper for acceptance.

The reviewers and I were impressed by several key strengths. The work addresses a clear gap in the literature, as PROFIT is one of the first optimizers tailored for fine-tuning already converged models. Its core idea of framing fine-tuning as temporal multi-task learning and using gradient orthogonalization to mitigate catastrophic forgetting is both simple and effective. A major advantage is its ease of integration—it's a "plug-and-play" wrapper that doesn't require architectural changes, making it highly accessible to practitioners. The authors back up their claims with extensive and compelling experiments across a diverse range of applications, from image classification and VQA to large-scale motion prediction, with the Waymo results being particularly noteworthy.

During the review process, some valid concerns were raised. One key point was the "proximal fine-tuning" assumption and how to determine if it holds. The authors have convincingly addressed this by demonstrating that a simple warm-up phase can adapt the method for non-proximal settings, and by highlighting numerous practical scenarios where this assumption is naturally met. Another important point was the lack of direct comparisons to methods like L2 regularization or continual learning approaches such as "Learning without Forgetting." While this is a valid critique, it does not diminish the core contribution of the paper. I strongly encourage the authors to include these comparisons and further discussion in their camera-ready version to provide a more complete picture for the community.

Overall, PROFIT presents a valuable, well-motivated, and empirically validated contribution that will be of great interest to the machine learning community. Its simplicity and effectiveness make it a promising tool for practitioners. This work is a solid addition to the NeurIPS program.